# Unraveling the hidden temporal range of fast β₂-adrenergic receptor mobility by time-resolved fluorescence

Ashwin Balakrishnan [1,4], Katherina Hemmen [1,4], Susobhan Choudhury[1], Jan-Hagen Krohn [1], Kerstin Jansen[1], Mike Friedrich[1], Gerti Beliu [1], Markus Sauer [1,2], Martin J. Lohse [3✉] & Katrin G. Heinze [1✉]

G-protein-coupled receptors (GPCRs) are hypothesized to possess molecular mobility over a wide temporal range. Until now the temporal range has not been fully accessible due to the crucially limited temporal range of available methods. This in turn, may lead relevant dynamic constants to remain masked. Here, we expand this dynamic range by combining fluorescent techniques using a spot confocal setup. We decipher mobility constants of β₂-adrenergic receptor over a wide time range (nanosecond to second). Particularly, a translational mobility (10 μm²/s), one order of magnitude faster than membrane associated lateral mobility that explains membrane protein turnover and suggests a wider picture of the GPCR availability on the plasma membrane. And a so far elusive rotational mobility (1-200 μs) which depicts a previously overlooked dynamic component that, despite all complexity, behaves largely as predicted by the Saffman-Delbrück model.

[1] Rudolf Virchow Center for Integrative and Translational Bioimaging, University of Würzburg, Josef-Schneider-Str. 2, 97080 Würzburg, Germany. [2] Department of Biotechnology and Biophysics, Biocenter, University of Würzburg, Am Hubland, 97074 Würzburg, Germany. [3] Max Delbrück Center for Molecular Medicine, Robert-Rössle-Straße 10, 13125 Berlin, Germany. [4]These authors contributed equally: Ashwin Balakrishnan, Katherina Hemmen. ✉email: lohse@toxi.uni-wuerzburg.de; katrin.heinze@virchow.uni-wuerzburg.de

G-protein-coupled receptors (GPCRs) belong to the largest family of cell surface receptors and are prime targets for therapeutic drugs[1]. They act as signal transducers converting ligand binding to a limited number of downstream effects. Being membrane proteins, they are bound to membranes that are transient, fluidic and highly ordered interfaces. Receptor diffusion on the membrane drives and governs cellular and downstream signaling. Molecular mobility or diffusion is key to elucidate the interplay of membrane proteins and the biophysics of the plasma membrane[2,3]. Understanding dynamic changes of GPCRs in terms of mobility would be crucial to understand how they are localized and how they function in intact cells.

When it comes to live-cell experiments, fluorescence techniques prevail and allow tackling inter- and intramolecular as well as diffusion dynamics[4–6] on various time scales with high spatial and temporal resolution. A prominently used fluorescence method for mobility determination and understanding oligomerization states is single particle tracking (SPT)[7]. SPT studies on various GPCRs have visualized mobility in the ~0.1–0.01 $\mu m^2 s^{-1}$ range with the existence of multiple populations that may be dependent on ligand binding[5,8–11]; they also point at distinct oligomerization states that receptors may display[8]. Unfortunately, every (fluorescence) approach for observing dynamics represents a compromise concerning temporal resolution and its range of time scales covered, field of view and throughput, spatial resolution in 2D and 3D, as well as sample preparation and choice of labels. Thus, SPT has given important insights into GPCR dynamics, however, the upper limit for diffusion coefficients "visible" by SPT is typically ~5 $\mu m^2 s^{-1}$. SPT is "blind" for faster kinetics of translational or rotational mobility[12,13]. Therefore, it is often necessary to use different, complementary approaches to gain a complete picture of a complex spatio-temporal setting. SPT and fluorescence correlation spectroscopy (FCS) can provide such complementary information[14]. In contrast, FCS[15] provides a faster, statistical analysis of photon bursts and can resolve diffusion dynamics in the lower $\mu$s to s scale even in live cells[16]. Time-resolved anisotropy (TRA) has been a useful tool to measure the even faster rotational diffusion and thus gives an idea of the microviscosity of the local environment in a wide range of systems[17,18], whereas oligomerization states have been analyzed mostly by FRET and other similar methods[19,20]. Both FCS and TRA have been employed in studies to determine molecular mobility of GPCRs before[21–23] but have a potential gap in the temporal spectrum around the 0.1–1 $\mu$s range. The polarization resolved FCS with full correlation (fullFCS) approach on the other hand covers the whole temporal spectrum from ns to s[24,25], closes the gap between TRA and FCS and thus lowers the risks to miss dynamic constants.

Here, we evaluated the diffusion dynamics of several fluorescently labeled $\beta_2$-adrenergic receptor ($\beta_2$-AR) constructs in live cells using FCS, TRA, and fullFCS in one setup. Thus, we were able to monitor receptor dynamics from the nanosecond to the second range, with and without ligand stimulation. We revisited previously reported dynamics to link their origin to function as a prerequisite to better understand downstream signaling. Our experiments further revealed previously hidden $\beta_2$-AR mobility that largely agree to the Saffman–Delbrück model despite previously raised doubts[23].

## Results

**Receptor constructs and experimental design**. Fluorescence observations of GPCRs in live cells (Fig. 1a) require their labeling, and both, label position and size, might have an influence on the mobility of the final construct. Therefore, we evaluated and compared different fluorescent labels and labeling strategies to ensure minimal impact on function, and we determined translational and rotational diffusion of the receptor before and after ligand binding (agonists, partial agonists, and inverse agonists, Fig. 1b). Thus, we devised $\beta_2$-AR constructs with different tagging strategies at different sites within the receptor, and with different label sizes (Fig. 1a, lower part). We compared four $\beta_2$-AR constructs with (i) EGFP conjugated to the N-terminus (NT), (ii) EGFP inserted into the intracellular loop-3 (IL3) at residue Q250, (iii) SNAP tag conjugated to the N-terminus (S)[9], which was labeled with an organic fluorophore, and (iv) genetic code expansion replacing the alanine residue 186 in the extracellular loop 2 with the unnatural amino acid *trans*-cyclooctene via amber suppression technology (TCO*lysine, $\beta_2$-AR$^{A186TCO}$) and using biorthogonal click chemistry for site-specific labeling with a small organic tetrazine-dye (TAG)[26,27]. Thus, the label sizes vary between 27 kDa for EGFP, 20 kDa for the SNAP tag and just ≈1 kDa for the Me-Tet-ATTO488 clicked to the TCO residue. All constructs were tested for their activity by measuring receptor-induced cAMP concentration[28] after transfection in CHO-K1 cells for NT and IL3 and in HEK-293T cells for TAG (Supplementary Fig. 1, Supplementary Table 1) and showed a similar trend in functional behavior compared to the untagged $\beta_2$-AR.

**$\beta_2$-AR exhibits translational diffusion on two different time scales**. We then studied translational diffusion, the lateral movement of the receptor by FCS in live cells. We performed in vivo measurements of the four different constructs on our custom-built two-channel confocal FCS setup (Supplementary Fig. 2) and auto-correlated the sum of the two channels (Fig. 2a). The obtained FCS curves were fitted under consideration of different diffusion models. We started with a superposition of a single two-dimensional translational diffusion coefficient ($D_{slow}$) and an additional relaxation time $\tau_T$ in the $\mu$s-time range (reflecting EGFP photophysics[29,30]) (2DT model). At a first glance it is already obvious that the simple 2DT model fails. Adding a second exponential relaxation time (2D2T) did not improve the data fit sufficiently (adj. $R^2$ of 0.978 and strong deviations in the weighted residuals towards the end in Supplementary Fig. 3b top row). Only the introduction of an additional diffusion component led to a suitable model: A second translational diffusion coefficient, although yielded a similar adj. $R^2$ of 0.975 (Supplementary Fig. 3b mid and bottom) and the weighted residuals were flat (Fig. 2a, top). The additional diffusion component that we introduced has been previously interpreted as "photophysics related"[31]. Therefore, we aimed to validate that this component arises from diffusion. One straightforward method is to simply change the pinhole size, as this increases the actual size of the probed FCS volume element. As the diffusion time derived from FCS measurements represents the average residence time of the molecular species within the focal volume, the respective decay of such a component should be directly related to the volume size, while photophysical processes can be assumed as largely independent of the size of the observation volume. When doubling the pinhole size, we found an increase in relaxation time for both components, confirming the assumption that they are diffusion-driven (i.e. both the slow and the fast component, Supplementary Fig. 4). Thus, we identified the 2D3DT model as the most likely one with a fast diffusion component (Fig. 2a; $\tau_{D2} = 0.75 \pm 0.10$ ms, light gray shaded area with a fraction of 0.28), a slower diffusion component ($\tau_{D1} = 68.8 \pm 1.1$ ms), and a photophysics component of EGFP with a relaxation time constant of 5 $\mu$s (gray-shaded area).

To further delineate this fast diffusion component, we screened the data from all other constructs (Fig. 2b, c). Consistently, over all four evaluated constructs two diffusion components were

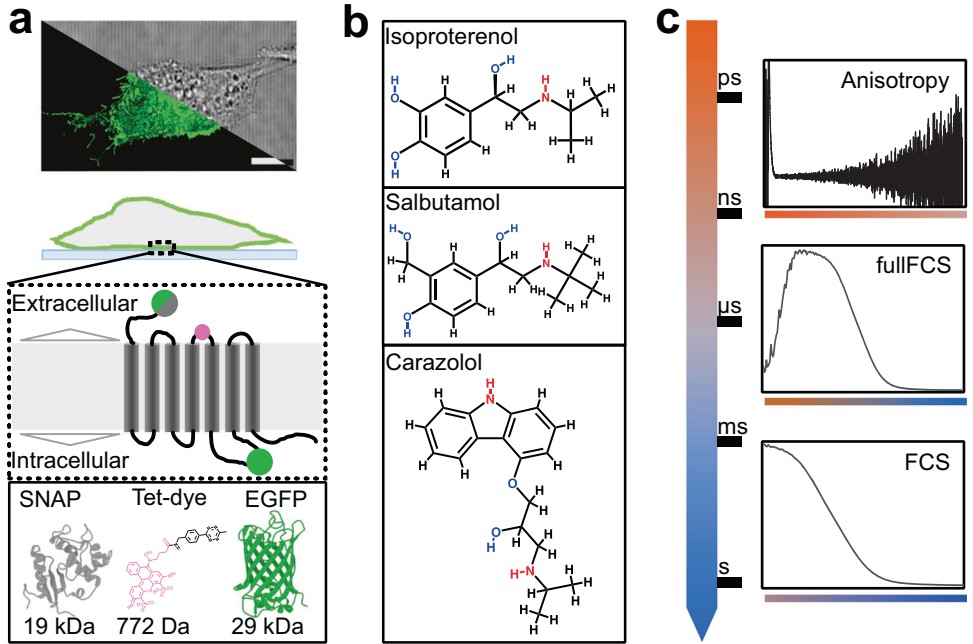

**Fig. 1 Fluorescence spectroscopy resolves membrane receptor mobility. a** CHO-K1 cells were grown on glass coverslips (top, fluorescent confocal maximum intensity projection vs. brightfield illumination) and transiently transfected with four different β₂-AR constructs (middle) carrying three differently sized fluorophores (bottom): N-terminal EGFP or SNAPtag, EGFP at the intracellular loop 3 or Me-Tet-ATTO488 coupled to the unnatural amino acid *trans*-cycloctene (TCO) at position 186 (β₂-AR$^{A186TCO}$). **b** Structures of agonist (isoproterenol, ISO), partial agonist (salbutamol, SAL) or inverse agonist (carazolol, CAR) used in this study to compare β₂-AR diffusion dynamics upon their addition to probe for possible changes in oligomerization or protein clustering. **c** The combination of three different time resolved spectroscopy experiments with overlapping temporal resolution allows to probe the translational and rotational dynamics in the range from ps to sec; Time-resolved anisotropy (TRA, top) is sensitive to rotation in the ps to the ns range, Fluorescence correlation spectroscopy (FCS, bottom) probes dynamics in the range from μs to s, while continuous-wave FCS (fullFCS, middle) covers a broad range from ps to s.

found with $D_{fast}$ in the range from 3 to 20 μm²/s and $D_{slow}$ in the range from 0.05 to 0.13 μm²/s (Fig. 2b). The fit results are summarized in Supplementary Table 2. The EGFP constructs (NT, IL3) and the TAG construct, had a larger fast diffusion fraction of 0.33, 0.31, and 0.27, respectively. The only potential exception was the $D_{fast}$ fraction of the S construct which exhibits only 0.07. As a comparison, we also tested equally labeled α₂A-AR constructs expressed in CHO-K1 cells and NT in HEK293T cells and found similar distributions of diffusion coefficients (Supplementary Figs. 5, 6 and Supplementary Table 2). Of note, the organic fluorophores used to label constructs S (SNAP-Surface Alexa Fluor 488) and TAG (Me-Tet-ATTO488) were chosen such that they had the same excitation and emission spectral range as EGFP. These two labels show typical triplet blinking of AF488[32] and ATTO488[33], respectively, at ~10 μs. SNAP-Surface Alexa Fluor 488 has been shown to be membrane impermeable[34,35], therefore fluorescent signal from inside the cell would be reduced in comparison with EGFP. Me-Tet-ATTO488 increases its brightness 20× upon binding to its substrate TCO[27] which would give 400× more autocorrelation signal than unbound Me-Tet-ATTO488 effectively highlighting only the fluorescent receptors.

As we consistently found a fast and a slow diffusion component for all four constructs, we tested several hypotheses to decipher their origin.

As we routinely perform FCS measurements in the basal membrane (plasma membrane side adhered to the coverslip), we wanted to test whether the GPCRs show a comparable diffusion behavior on the apical membrane (plasma membrane side away from the coverslip) to avoid systematic bias. Although nearly twice as many receptors were found at the apical compared to the basal membrane (Supplementary Fig. 7a) (as reported earlier[36]),

no difference in receptor mobility was observed. Both diffusion coefficients $D_{fast}$ and $D_{slow}$ as well as the fraction of fast diffusion molecules were indistinguishable from basal membrane measurements (Supplementary Fig. 7b, c). Thus, we can assume that the (tight) attachment of a cell to the glass surface does not significantly change the diffusion behavior of the membrane receptors studied here. There was no correlation between the receptor concentration and mobility apparent (Supplementary Fig. 7d), which suggests unaltered receptor behavior in terms of clustering (receptor concentration was calculated for all constructs based on the focal volume element and the number of molecules in focus from the 2D3DT fits).

The confocal volume has an axial elongation of ~2 μm (derived from calibration with freely diffusing AF488 dye) while the plasma membrane of an eukaryotic cell is only around 10 nm in thickness[37]. Consequently, we also capture fluorescence from (labeled) molecules inside the cell[29]. Here, our experimental setting provides an inherent control and a strong hint that intracellular mobility is indeed the source of the additional fast diffusion component. The organic fluorophore used for the S construct is membrane impermeable so that only β₂-AR molecules localized at the cell surface were available for labeling. We saw indeed a reduction of the rapid component (Fig. 2c) for the SNAP label. Although one might argue that the rapid mobility of the SNAP label might arise from outside the cell since it is membrane impermeable, washing usually removes any unbound dye. The expected diffusion of free SNAP label would be in the order of ~400 μm² s⁻¹ [38,39], which is an order of magnitude higher than the observed value. In addition, we performed confocal imaging of CHO-K1 cells transfected with SNAP label to make sure that fluorescence arises only from the cell

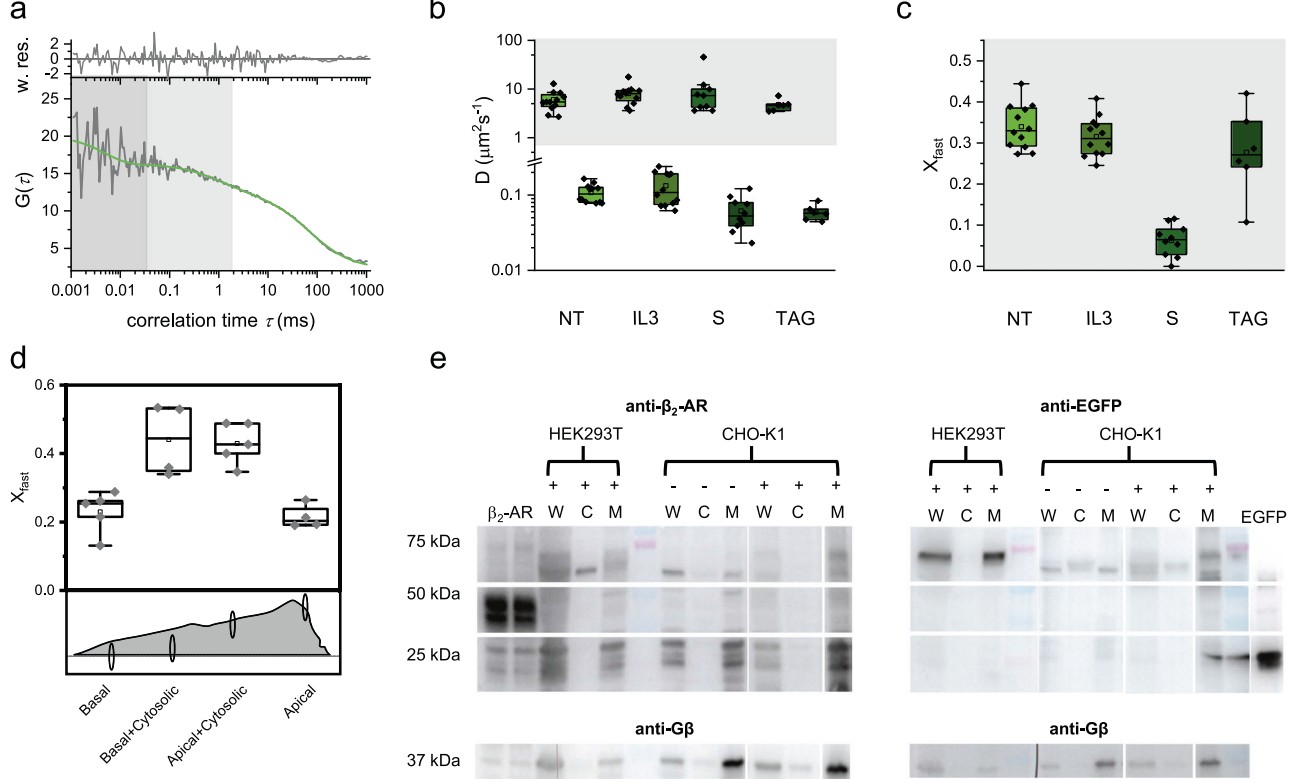

**Fig. 2 FCS data reveal β₂-AR diffusion on two different time scales. a** FCS curve derived from the basal membrane of a CHO-K1 cell transiently transfected with N-terminal-EGFP-β₂-AR (NT). The curve was fitted with the model 2D3DT (Eq. (4)) containing two diffusion times ($t_{D1}$ = 68.7 ± 1.13 ms, white and $t_{D2}$ = 0.75 ± 0.10 ms, light gray) and a relaxation time ($t_T$ = 5.00 ± 5.68 μs, gray). The weighted residuals are shown on top. Fit results are mean ± s.e.m. **b** Diffusion constants, $D_{fast}$ (light gray area) and $D_{slow}$, calculated from the corresponding diffusion times $t_{D1}$ and $t_{D2}$, respectively (Eq. (5)) of NT (light green, 6.08 ± 2.80 and 0.10 ± 0.02 μm² s⁻¹ for $n$ = 12), IL3 (green, 8.09 ± 3.65 and 0.13 ± 0.06 μm² s⁻¹ for $n$ = 12), S (meadow green, 11.0 ± 13.2 and 0.06 ± 0.03 μm² s⁻¹ for $n$ = 10) and TAG (dark green, 4.74 ± 1.36 and 0.06 ± 0.01 μm² s⁻¹ for $n$ = 6). **c** Fraction of molecules exhibiting fast diffusion for the same constructs as in (**b**). NT = 0.33 ± 0.05; IL3 = 0.31 ± 0.04; S = 0.07 ± 0.03 and TAG = 0.27 ± 0.10. Results for $α_{2A}$-AR are shown in Supplementary Fig. 5. Data are mean ± s.d. **d** Fraction of molecules exhibiting fast diffusion for NT at different positions on and partly on the plasma membrane. The lower cartoon represents the respective focal positions. The focal volume and the dimension of the cell are drawn to scale. The dimensions of the focal volume were 2.72 μm × 0.46 μm and the dimensions of the representative cell were 9 μm × 40 μm. Basal 0.22 ± 0.06; Basal + Cytosolic 0.44 ± 0.10; Apical + Cytosolic 0.42 ± 0.06; Apical 0.21 ± 0.03. Data are mean ± s.d. Diffusion constants for all the cell are shown in Supplementary Fig. 9. **e** Western Blot of untransfected (−) CHO-K1 and NT transfected (+) HEK293T and CHO-K1 cells. For both samples, the whole cell lysate (W), the separated cytosol (C) and membrane (M) fractions were loaded (see the section "Methods" for details). As a positive control for NT, HEK293T transfected with untagged β₂-AR ("β₂-AR"), and for EGFP, purified EGFP ("EGFP") were used. As loading control the same gels were probed for Gβ after stripping. The full blots along with the controls can be seen in Supplementary Fig. 10. Blots for S construct can be seen in Supplementary Fig. 11.

(Supplementary Fig. 8). To follow up our hypothesis of β₂-AR partially being localized intracellularly, we performed axial scanning through a single cell. The position of the focus (centered on the membrane or decentered) changed the observed fraction of fast versus slow diffusing receptors; fast diffusing receptors were found to be more pronounced in the decentered position (i.e. when the observation volume reaches more into the cytoplasm). In return, the slow diffusing species was found to be more pronounced for the focus position centered on the membrane, regardless of whether basal or apical membrane (Fig. 2d). The characteristic diffusion coefficients remained constant (Supplementary Fig. 9).

To elucidate the source of this intracellular signal further, we performed Western blotting of the NT construct using both anti-EGFP and anti β₂-AR antibodies (Fig. 2e, Supplementary Fig. 10). Here, we probed the whole cell lysate as well as the separate fraction of membrane and cytosol. As positive control we used the whole cell lysates of transiently transfected HEK293T cells expressing β₂-AR[40] as well as purified GFP. Our full-length NT has a molecular weight of ~74 kDa. Free EGFP (if spliced from the construct) would result in a band of ~27 kDa. The Western

blot clearly shows the absence of NT or free EGFP in the cytosol, i.e. the whole protein is present in the membrane fraction. We performed similar blots of the S construct but with purified SNAP-tag as a positive control instead of GFP. Similar results were obtained in the case of construct S (Supplementary Fig. 11). Thus, the intracellular diffusion most likely results from β₂-AR located on intracellular membranes.

**β₂-adrenergic receptors show fast rotational correlation with dependency on the fluorophore labeling position.** Next, we evaluated the rotational motion of the β₂-AR in the cell membrane by fluorescence anisotropy. For polarized excitation and polarization-dependent data collection in our custom-built setup, the fluorescence emission was split into two detection channels, one parallel to the plane of polarization as the exciting laser pulse (called VV) and the other perpendicular with respect to the plane of polarization of the incident laser pulse (called VH). For analysis, we reconstructed the photon arrival time histograms of both channels and analyzed the time-resolved fluorescence anisotropy (TRA; Eqs. (7) and (8)). The intensity decays $I_{VV}$ and $I_{VH}$ were

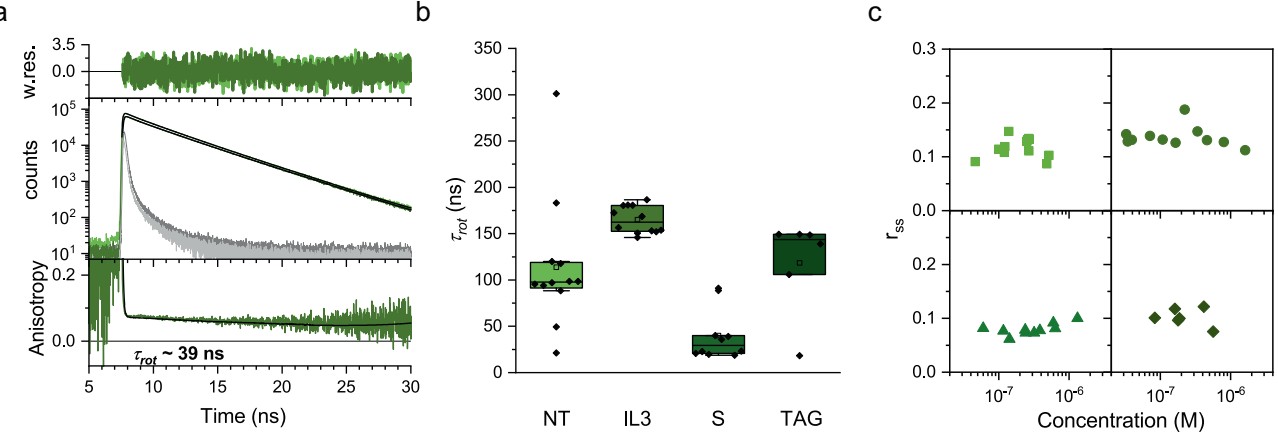

**Fig. 3 Time-resolved anisotropy show fast rotational correlation dependent on fluorophore position. a** Photon arrival time histograms of vertically (green) and horizontally (light green) polarized photons arising from a CHO-K1 cell transiently transfected with N-terminal-SNAP-$\beta_2$-AR conjugated to Alexa Fluor 488 dye. The instrument response functions (IRF) for vertical and horizontal polarization are shown in gray and dark gray, respectively. The histograms were fit globally with Eqs. (7) and (8) to obtain a slow rotational correlation time ($\tau_{rot}$) of 39 ns. The fits are shown in black. The bottom panel gives the reconstructed anisotropy decay and the corresponding fit. Weighted residuals corresponding to the global fits are shown on top. **b** Rotational correlation times of all $\beta_2$-AR constructs, naming and color scheme as in Fig. 2. NT shows 113.7 ± 70.5 ns; IL3 shows 165.1 ± 14.6 ns; S shows 40.0 ± 27.5 ns and TAG shows 118.3 ± 51.8 ns. Please note that from biexponential rotational correlation time fitting, the slower $\tau_{rot}$ was used. Data are mean ± s.d. **c** Steady-state anisotropy ($r_{ss}$) is largely independent from receptor density within the measured concentration range of all $\beta_2$-AR constructs. Squares represent NT, circles represent IL3, triangles represent S and diamonds represent TAG.

jointly fitted to a multiexponential fluorescence lifetime model and a biexponential rotational correlation time, $\tau_{rot}$. Only the longer rotational correlation time was considered for further results as it is associated with the receptor dynamics and the shorter component arises from free rotation of the fluorophore head group. For small organic dyes, rotational correlation times arising from the freedom of movement of the conjugated dye molecule can be differentiated from that arising from the bulkier protein. However, in some cases we required a triexponential rotational correlation fit for TAG which could be arising from free dye (Supplementary Table 3 and see the "Methods" section). The fundamental anisotropy of EGFP and both SNAP tag substrate and Me-Tet-ATTO488 was fixed to 0.38 according to the value given in the literature[41,42]. Figure 3a shows exemplary data for the $I_{VV}$ (dark green) and $I_{VH}$ (light green) overlaid with the fit (black) from a measurement of the S construct. Respective instrument response functions (IRF) are shown in dark (for VV) and light gray (for VH). The bottom panel shows the recalculated TRA with a slow $\tau_{rot}$ of 39 ns. Here, clearly, the initial fast decay due to the fluorophore rotation (<1 ns time range) can be differentiated from the slower depolarization due to the receptor motion (>20 ns time range). At the first glance, the rotational correlation times (Fig. 3b, Supplementary Fig. 6c, Supplementary Table 3) ranging between 20 and 300 ns seem to be different for the four constructs, however, careful sampling of the $\chi^2$ surface[43] reveals no significant differences as observed in the case of translational diffusion coefficients. Similar results were observed for the $\alpha_{2A}$-AR constructs (NT-A and S-A, Supplementary Fig. 12a).

**Receptor mobility seems independent from receptor density.** Since FCS and TRA were obtained from the same intensity traces, the TRA data can be directly related to the FCS data. To test receptor oligomerization, we plot the receptor density (or concentration) against the steady-state anisotropy ($r_{ss}$) as described previously[43,44]. As the oligomerization probability is higher for dense receptor distributions, $r_{ss}$ is expected to decrease with increasing receptor density in the case of oligomerization that leads to FRET between same fluorophores (homoFRET) and thus

increased depolarization. Figure 3c shows no correlation between the receptor density and $r_{ss}$ for all constructs in the concentration range measured (20 nM–5 μM). Similar results were observed for the $\alpha_{2A}$-AR constructs (Supplementary Fig. 12b). It should be noted that the $r_{ss}$ values in our data are ~0.1 which denotes that there should be homoFRET inherently present at all the concentrations that we measured indicating the presence of basal oligomerization. If density independent oligomerization mechanisms[45,46] are at play they could not be accessed here. To take a closer look at the potential influence of receptor density, all obtained mobility parameters ($D_{fast}$, $D_{slow}$, $x_{fast}$, $\tau_{rot}$) were mapped against the concentration of the respective sample (Supplementary Fig. 13) and seem to be largely concentration-independent, thus indicating absence of receptor clustering.

**fullFCS reveals an additional rotational correlation time of $\beta_2$-AR.** With two translational diffusion times derived from FCS, and only one receptor associated rotational diffusion time from anisotropy experiments, we wondered whether there is an additional hidden component that we might have missed due to unfavorable temporal resolution limits of the two approaches. To reveal any potential slower rotational correlation times, we therefore performed fullFCS for our NT and S constructs in live cells to uncover diffusion and relaxation dynamics down from nanoseconds all the way to seconds (Fig. 4). NT and S were chosen as representative for receptors bound to a fluorescent protein and organic fluorophore, respectively. As fullFCS requires long acquisition times in live cells to exhibit reasonable resolution in the ns range, we adapted an approach that allows automated FCS data analysis with efficient rejection of corrupted parts of the signal as previously published by Ries et al.[47]. This enabled us to apply fullFCS in live cells despite the large photon statistics required for fullFCS. Here, the photon trace is sliced into smaller pieces with each piece individually correlated. Only pieces that fulfill our threshold criteria were further considered for final data analysis (Supplementary Fig. 14). This way we could manage out-of-focus movements of cells without discarding entire experiments. In brief, 18 correlation curves based on five different slices and the full trace per cell were fitted globally with shared diffusion

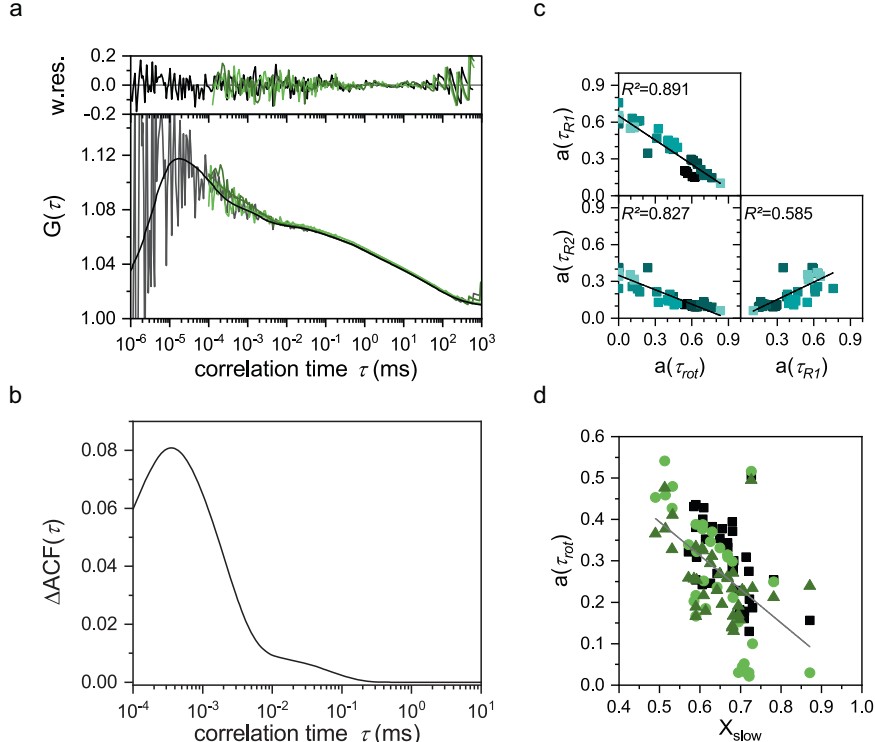

**Fig. 4 Continuous wave FCS hints at additional slow rotation. a** Exemplary fullFCS measurement from a CHO-K1 cell transfected with N-terminal-EGFP-$\beta_2$-AR. For the data sliced in 5 s pieces a difference in amplitude between the $CCF_{VV-VH}$ (black) and $ACF_{VH}$ (light green) compared $ACF_{VV}$ (green) can be seen. The data was globally fitted with Eq. (10). Weighted residues are shown on top. **b** The absolute difference in amplitude between $ACF_{VV}$ and $ACF_{VH}$ calculated from the fit parameter (Eq. (12)). **c** For the $ACF_{VH}$-curves, an increase in amplitude of $\tau_{rot}$ leads to decreasing amount of the two $\tau_R$ in the μs range. Both $\tau_{R1}$ and $\tau_{R2}$ are positively correlated. Each color represents the measurement from one cell ($n = 7$). $\tau_{rot} = 99 \pm 52$ ns, $\tau_{R1} = 1.90 \pm 0.45$ μs, $\tau_{R2} = 128 \pm 39$ μs. Data are mean ± s.d. **d** An increased amount of slow, membrane diffusion, $x_{slow}$, is related to a decreasing amplitude of $\tau_{rot}$. The color code is same as in (**a**). Results for $CCF_{VV-VH}$ and $ACF_{VV}$ and for the N-terminal-SNAP-$\beta_2$AR conjugated to AF488 are summarized in Supplementary Figs. 16c–f and S17.

and relaxation times (Fig. 4a) as described in Eq. (10). Figure 4a shows the exemplary results of an NT measurement. The cross-correlation between the "VV" and "VH" channel ($CCF_{VV-VH}$, black fit, transparent data) extends into the ns time range and shows also photon antibunching[48] ~3 ns, related to the excited state lifetime of EGFP. In contrast, both auto-correlations ($ACF_{VV}$, dark green, and $ACF_{VH}$, light green) only covered the range up to 100 ns. For the NT constructs, $ACF_{VH}$, $ACF_{VV}$, and $CCF_{VV-VH}$ showed three additional exponential relaxation times next to the two translational diffusion times determined above. The fastest relaxation time ($99 \pm 52$ ns) agrees well with the rotational correlation time from TRA ($113 \pm 70$ ns), thus we named this one $\tau_{rot}$, the two other relaxation times $\tau_{R1}$ ($1.9 \pm 0.45$ μs) and $\tau_{R2}$ ($128 \pm 39$ μs) are in the μs range. The faster relaxation time $\tau_{R1}$ maybe the typical time constant for EGFP photophysics[30]. All fit results are summarized in Supplementary Table 4.

A characteristic feature of rotational motions is their influence on the depolarization of the emitted light after polarization-specific excitation of a fluorophore[25]: Depending upon the rotational speed, the signal was different in our two correlation channels and we observed a small difference in the amplitude of the $ACF_{VV}$ and $ACF_{VH}$ curves in the 100 ns–10 μs range. Each set of fitted $ACF_{VH}$, $ACF_{VV}$, and $CCF_{VV-VH}$ was then compared to see changes in amplitude of the fitted relaxation terms. To visualize this difference, we calculated the absolute difference Δ between the fit results of the $ACF_{VV}$ and $ACF_{VH}$ curves (Fig. 4b, Supplementary Fig. 15). There was a minor difference at ~50 μs, and a second prominent peak at ~500 ns.

To test whether one of the additional relaxation times $\tau_{R1}$ and $\tau_{R2}$ might reflect a rotational correlation time that had been missed so far, we plotted their fraction versus the fraction of the two other relaxation times (Fig. 4c) and then correlated the amplitude of the identified $\tau_{rot}$ with the fraction of slow diffusion (Fig. 4d). Interestingly, the amplitude of $\tau_{rot}$ of the $ACF_{VH}$ was negatively correlated with both $\tau_{R1}$ and $\tau_{R2}$ with correlation coefficients $R^2$ of 0.89 and 0.82, respectively, while $\tau_{R1}$ and $\tau_{R2}$ were slightly positively correlated with $R^2$ of 0.58 (Fig. 4c). The analysis of $CCF_{VV-VH}$ and $ACF_{VV}$ amplitudes shows similar results (Supplementary Fig. 16a, b). Most important is that the amplitude of $\tau_{rot}$ decreased with an increased amount of slow diffusion in the sample, indicating a relation between the fast diffusion and fast rotation, whereas the slower relaxation times, might be associated with the slower diffusion in the membrane. Our analysis thus hints to an additional (slow) rotational diffusion in the 1–200 μs time range which would be the missing piece needed to resolve contradictions in applying the Saffman–Delbrück model[49] to describe GPCR dynamics. Similar results were obtained for the S construct (Supplementary Figs. 16c–f and 17), however, another additional relaxation time was required for proper fitting of the curves (Supplementary Table 4).

**Ligand stimulation of $\beta_2$-AR affects diffusion constants.** Finally, we evaluated the effects of agonists and inverse agonists on the receptor's translational and rotational mobility. We performed stimulation with different ligands (Fig. 1b), isoproterenol

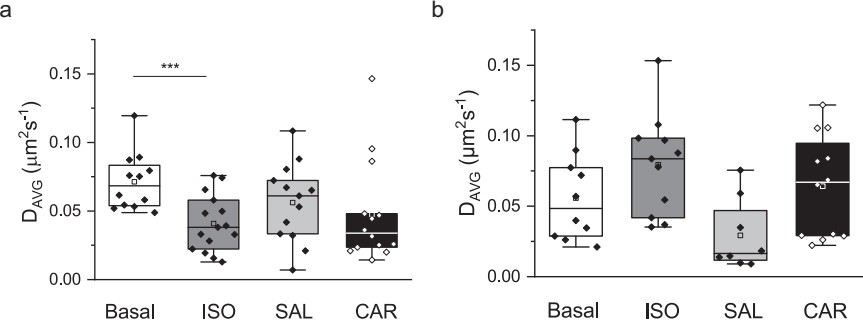

**Fig. 5 Ligand stimulation affects diffusion constants. a** Ligand effects on membrane diffusion of NT expressed in CHO-K1 cells. The cells were incubated for 5 min with each of the ligands (ISO and CAR at 1 μM and SAL at 2.4 μM) prior to measurement. Average diffusion constants, $D_{AVG}$, weighted over the corresponding species fractions (Eq. (5)), are shown: Basal untreated state (white) $0.07 \pm 0.04\ \mu m^2\ s^{-1}$, ISO (dark gray) $0.04 \pm 0.02\ \mu m^2\ s^{-1}$, SAL (light gray) $0.06 \pm 0.03\ \mu m^2\ s^{-1}$ and CAR (black) $0.05 \pm 0.04\ \mu m^2\ s^{-1}$. *** is $P < 0.001$. **b** Ligand effects on membrane diffusion of S expressed in CHO-K1 cells. The incubation was the same as NT. Average diffusion constants, $D_{AVG}$ for the basal and ligand treated states are shown: Basal untreated state $0.05 \pm 0.02\ \mu m^2\ s^{-1}$, ISO $0.08 \pm 0.03\ \mu m^2\ s^{-1}$, SAL $0.03 \pm 0.02\ \mu m^2\ s^{-1}$, and CAR $0.06 \pm 0.03\ \mu m^2\ s^{-1}$.

(ISO, agonist), salbutamol (SAL, partial agonist) and carazolol (CAR; inverse agonist) (Fig. 5) of $\beta_2$-AR for 5 min prior to measurement. Next, we performed translational and rotational diffusion analyses as described above for the unstimulated constructs. Similar to untreated cells, the treated NT construct required a fit model with two diffusion times (Supplementary Table 5). To account for the cell-to-cell variation, we calculated an average diffusion coefficient $D_{AVG}$, weighted by the species fractions (Eq. (6)) (individual values are shown in Supplementary Figs. 18a–c and 19a–c). Figure 5a shows a significant reduction in $D_{AVG}$ of almost an order of magnitude in the ISO-treated cells compared to the untreated cells. An insignificant reduction was seen for the partial agonist SAL, whereas $D_{AVG}$ of the inverse agonist CAR showed a broad distribution with a significant decrease in median and mean values. An additional phenomenon that could lead to this decrease in diffusion constant upon activation is initiation of internalization. In order to understand if internalization of receptors was playing a role we treated cells expressing NT with pitstop2. Pitstop2 has been shown to be an inhibitor of clathrin dependent[50] and clathrin independent endocytosis[51]. Supplementary Fig. 20 shows a decrease in the slow diffusion constant with pitstop2-treated cells similar to ISO treatment. On the other hand, for ligand stimulation of the S construct (Fig. 5b) no significant change in $D_{AVG}$ was observed (Supplementary Table 5) which is in agreement with previously published work[8].

At the same time, the distribution of the rotational correlation times seemed to decrease to 20–100 ns for NT (i.e. the receptors rotated faster) after ligand stimulation for all three ligands (Supplementary Fig. 18d). In the case of S, ISO stimulation decreases the distribution of the rotational correlation times to around 20 ns whereas the range was not significantly different for stimulation with partial agonist and inverse agonist (Supplementary Fig. 19d). All constructs exhibited biexponential rotational correlation times with the exception of CAR treated SNAP cells where a bi- and triexponential fit were used and the appropriate fit was chosen based on (i) visual inspection of the weighted residuals (did the deviations in weighted residuals decrease in the tri-exponential fit compared to the bi-exponential fit) and (ii) a $\chi^2$-criterion (Supplementary Fig. 19e). We calculated the relative $\chi^2_{rel}$ ratio as $\chi^2_{bi}/\chi^2_{tri}$ and defined the $2\sigma$ threshold (95% confidence level, 1700 data points, 16–18 parameters) based on an F-test, to accept the tri-exponential fit if $\chi^2_{rel} < 1.016$. Fit results are summarized in Supplementary Table 6. In essence, ligand treatment does not interfere with the rotational mobility although there are shifts in the correlation times.

To explore the potential effects of receptor density, we mapped all mobility measures against the concentration (Supplementary Figs. 18a–c, f, g, 19a–c, g, h). For both NT and S, the density of the receptors does not seem to influence the translational diffusion dynamics (Supplementary Figs. 18a–c and 19a–c). Notably, the rotational correlation time of the receptor in NT (Supplementary Fig. 18f) are tightly clustered for the ligand stimulation while the untreated cells show a broad distribution. On the other hand the rotational correlation times are distributed broadly in the case of S (Supplementary Fig. 19g). The steady-state anisotropy ($r_{ss}$) stayed constant for both constructs (Supplementary Figs. 18e, g and 19f, h). These results show that there is no change in receptor oligomerization states upon ligand stimulation.

**Discussion**

In this work, we were able to show how ARs fulfill a variety of biological functions at different regions in the cell through a large range of $\beta_2$-AR mobility. In addition to the previously reported "classic" (relatively slow) membrane diffusion, we revealed an additional fast translational diffusion that likely originates from internalized receptor vesicles close to the plasma membrane by excluding all other common players. Moreover, additional rotational diffusion constants became accessible here by fullFCS which gives key insights into the diffusion dynamics of GPCRs and the proper models to link the individual mobility which would not have been accessible by the mostly used SPT techniques due to its narrower temporal range of accessible receptor dynamics. Thus, we pinpointed the so far elusive component of the μs-rotational diffusion of $\beta_2$-AR which is crucial to understand that $\beta_2$-AR, despite its complexity, fulfills the Saffman–Delbrück model. In addition, mobility upon ligand stimulation is in line with previous studies[9,52] and hints at basal oligomerization states in $\beta_2$-ARs. In Fig. 6 we summarize the revised model of $\beta_2$-AR mobility.

All $\beta_2$-AR and $\alpha_{2A}$-AR constructs we analyzed exhibit two different translational diffusion constants. The slow diffusion component we observed in the range from 0.05 to 0.13 μm²/s has been reported before using SPT on GPCRs as subpopulations undergoing different diffusion[9]. In our case the average of the subpopulations is measured. However, the fast diffusion component in the range from 3 to 15 μm²/s, has either been ignored or misattributed to photophysics[31] in the past. Here, we could not only show the existence of such fast GPCR mobility, but could also pinpoint its biological relevance, and exclude bias concerning antibody accessibility for basal versus apical membrane as

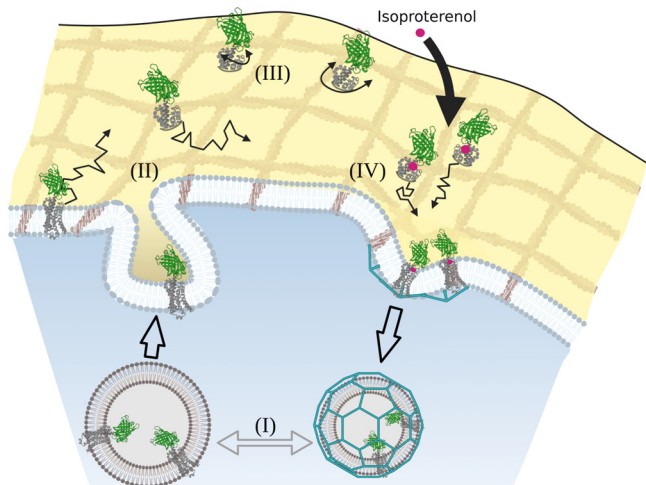

**Fig. 6 Revised model of β₂-AR dynamics.** Schematic of cellular cross-section indicating the mobility spectrum of fluorescently labeled β₂-ARs. (I) Fast translational diffusion originating from vesicle transport close to the membrane. (II) Slower "classic" β₂-AR membrane translational diffusion. (III) Rotational membrane diffusion of β₂-AR. (II) and (III) occur simultaneously in all receptors; here illustrated separately for simplicity. (IV) Decreased "classic" β₂-AR membrane translational diffusion upon activation by ligand binding (here: isoproterenol in magenta). The graphic was created with Biorender.com.

recently reported in some other cell settings[29,36,37]. The decreased fast component fraction shown by S (labeled with membrane impermeable dye) showed that the fast diffusion has intracellular origins. Western blotting confirmed the absence of soluble membrane receptors in the cytosol. A similar pattern of two different diffusion constants has been shown in some membrane-associated proteins using fluorescence recovery after photobleaching which was shown to be arising from lateral membrane and membrane-cytoplasmic exchange[53]. We assume a similar membrane–cytosol exchange of receptors via vesicles where on one hand newly produced receptors from the endoplasmic reticulum via golgi or recycled receptors are transported to the membrane and on the other hand receptors are internalized for degradation or recycling with the help of downstream proteins. Vesicles exhibit diffusion constants in the range of 2–40 μm²/s[54,55], which is in line with the fast diffusion component we determined. Taken together our results imply that the fast diffusion constant arises from intact membrane bound receptors on the cytoplasmic side, such as those associated with vesicular transport.

Next, we had a closer look at the rotational diffusion of our receptor constructs. On one hand, our correlation time derived from all four constructs (Fig. 3b) support previously reported findings of very fast rotational correlation times of 20–300 ns observed in other GPCRs[22,23]. On the other hand, this in consequence would mean that β₂-AR rotational diffusion does not support the Saffman–Delbrück approximation[49] for diffusion of membrane proteins, which predicts a rotational correlation time of 160 μs for adrenergic receptors (for a radius of ~2 nm and the obtained mean of $D_{slow}$). The Saffman–Delbrück approximation predicts the ratio of the lateral to rotational diffusion coefficient for a given transmembrane protein undergoing Brownian diffusion in a finite membrane space and a non-crowded environment. This large difference between expected and obtained experimental result caught our attention, as the main disadvantage of TRA is its limited temporal resolution dependent on the fluorescence lifetime of the fluorophore used[56]. Here, establishing fullFCS[25] in

live cells was the key to see diffusion dynamics from the nanosecond to the second range. Our fullFCS fits suggest μs rotational correlation times in the case of NT and S. Further additional rotational correlation times observed for S could represent free dye rotation[57] (or diffusion of unbound substrate) which is also consistent with our TRA fits. The discovery of previously hidden μs rotational correlation times in NT and S allows us to model the β₂-AR dynamics in accordance to the Saffman–Delbrück approximation which otherwise would—based on the previously reported data[22,23]—be violated. It is known that the Saffman–Delbrück model considers a rather ideal membrane, without the diverse membrane composition and crowding environment of a living cell[58,59]. Nevertheless, our results exhibit rotational correlation times in the range of 1–200 μs associated with $D_{slow}$ of 0.05–0.13 μm²/s.

Mobility measurements of all constructs lead us to understand their influence on the measurement itself. For the four used constructs (NT, S, IL3 and TAG) the translational diffusion behavior was quite robust; only the ratio between the two diffusion coefficients appeared significantly different for the S construct, most likely due to the labeling strategy with a membrane-impermeable fluorophore. In contrast to the translational diffusion, the rotational diffusion showed more pronounced dependencies on label type and position which is not easy to pinpoint to a specific source. Particularly the large size of fluorescent proteins, makes it impossible to decouple the motion of the fluorophore and the receptor. However, inserting the EGFP into an intracellular loop restricts its rotation to a minimum so that it is the favorable setting if using FPs as long as the receptor allows such an insertion without altering function. If possible, small, bright labels are favorable for time-resolved fluorescence spectroscopic approaches; labeling strategy and positions need always careful evaluation and thorough calibration.

After interrogating the different constructs we probed the effect of ligand interaction for the NT and S construct representing the two major labeling strategies. Our treatment of NT with isoproterenol shows a significant decrease in the slow diffusion constant while the changes upon carazolol and salbutamol treatment are non-significant here. The S construct does not show any significant changes, which is in accordance with a recent single particle tracking[9] and a FRET study[52]. This difference could be attributed to the inherent differences of NT and S. The S construct in comparison with NT has less fluorescent receptors as only the plasma membrane bound receptors are labeled in the case of S, whereas in the case of NT, receptors in the endoplasmic reticulum and those undergoing vesicular transport to and from the plasma membrane are also fluorescent. The basal receptor internalization during the period between labeling and measurement decreases the amount of plasma membrane bound fluorescent receptors in S. As soon as the receptors are activated, clathrin coated pit formation increases[60], and as a consequence internalization rate increases[61]. A decrease in the diffusion constant occurs as the ensemble of the fluorescent receptors being probed has more clusters in the clathrin coated pits. The number of fluorescent clusters in the case of S might be lower relative to NT since the plasma membrane bound fluorescent receptors are fewer in number. In the case of S upon agonist treatment, this reflects in the FCS measurement as the diffusion time of the ensemble of the plasma membrane bound receptors being not significantly different from its basal state. This would in turn lead to a decrease in $D_{slow}$ as shown by NT upon agonist activation, which gives us access to all receptors relative to S. Blocking internalization in the basal state had a similar effect where the diffusion constant decreased, which may be due to clusters accumulating on the plasma membrane which would normally have been internalized.

Another mechanism that could be at play in tandem with clustering and internalization leading to the decrease in the diffusion constant is the presence of micro- or nanodomains. Lipid molecules constituting nanodomains have been shown to possess lower diffusion constants relative to the surrounding lipid molecules[62] and a recent review sums up how protein diffusion might be influenced by nanodomains in combination with other factors[63]. Other factors such as interaction with downstream partners such as Gs complex and β-arrestin might also play a role although it is hard to discern this in terms of mobility alone. The main reason as theorized by the Saffman-Delbrück model[49] and its approximations[58,64] being the dependence of viscosity on mobility. A small change in the hydrodynamic radius of the molecule (~2 nm for a receptor to ~4 nm for the complex) would not show a considerable change in mobility. Another aspect to be considered is that the interaction time of receptor/G-protein is quite short lived due to high GTP level in cells leading to fast dissociation[5].

The rotational diffusion on the other hand seems to get faster upon ligand treatment, at least in the case of NT, as also reported for the FlAsH-tagged $\alpha_{2A}$-receptor[23]. However, since EGFP, AF488 and ATTO488 have a relatively short fluorescence lifetime (2.9–4.1 ns), the fluorescence intensity completely decays within 30–50 ns. This means that rotational correlation times >50 ns cannot be determined with confidence especially in our case where tumbling/rotation of the whole construct (60–90 kDa) is predicted to be in the µs as mentioned before. It has to be stressed that our mobility data does not signify any ligand-induced conformational changes as protein conformational changes usually do not affect translational or rotational mobility, the latter being determined by overall molecule size.

In addition to mobility, we probed changes in oligomerization states using steady state anisotropy ($r_{ss}$) and homoFRET[19]. Our $r_{ss}$ data at basal state, with its value of ~0.1 shows that homoFRET and hence oligomerization in the basal state occurs. Ligand-induction did not change this trend, signifying that the basal equilibrium of monomer vs. oligomer states is unchanged upon ligand activation as previously reported[8]. Overall, our results signify ligand-induced changes in receptor mobility caused by multiple mechanisms ranging from clustering, internalization, micro-/nanodomains and receptor/G-protein/β-arrestin interactions playing a combined role.

In this study, we overcame a blind spot on the map of GPCR dynamics in the µs range by using advanced time-resolved spectroscopic techniques including FCS and TRA in conjunction with fullFCS. Our data demonstrates the presence of different pools of GPCRs in terms of their mobility and function at different cellular loci that should be also considered in future investigations for other membrane associated proteins.

## Methods

**Plasmids.** Three different strategies of fluorescent tags were employed, EGFP, SNAP and unnatural amino acid. In the case of $\beta_2$-AR all three tags were incorporated and in the case of $\alpha_{2A}$-AR only EGFP and SNAP were included. EGFP was tagged in the $\beta_2$-AR in the N-terminal (NT) and in between the intracellular loop-3 (IL3) and in the $\alpha_{2A}$-AR in the N-terminal (NT-A). SNAP was tagged to the n-terminal of both $\beta_2$-AR (S) and $\alpha_{2A}$-AR (S-A). For Unnatural Amino Acid tagging a TAG codon was engineered at position 186, namely $\beta_2$-AR$^{A186TCO}$ (TAG). The plasmid for the expression of tRNA/tRNA-synthetase pair, pCMV tRNA$^{Pyl}$/NESPylRS$^{AF}$ was kindly provided by Prof. Edward Lemke[65]. All plasmids were constructed based on a pcDNA3 vector backbone with the receptor inserted in between the enzyme cleavage sites of XhoI and HindIII. All constructs except S and S_A were tested for their functionality using cAMP assay kit (Abcam, Cambridge, UK) (Supplementary Table 1). The functionality of S and S_A have been reported before[8,9].

**Functional assay using fluorimetric cAMP analysis.** cAMP response to concentration series of isoproterenol was performed using the commercial fluorimetric cAMP kit from abcam (ab138880). All cells transfected with different constructs

were treated with the respective concentration of isoproterenol (6 different concentrations from 100 pM to 10 µM) and incubated for 1 h after ligand addition. The cells were lysed post incubation and added to a 96-well plate coated with anti-cAMP antibodies and incubated for 10 min. After this, a solution of HRP-cAMP was added to all wells and incubated for the next 2 h with shaking. The wells were washed with a wash buffer after the incubation and a fluorophore solution capable of binding HRP was added to all wells and incubated for 10 min. Fluorescence was measured from the plate post incubation by exciting using 540/10 nm light and emission detected at 590/10 nm. The fluorescence intensity detected is proportional to the inverse of the cAMP concentration as determined by the preparation of standard curve with known cAMP concentrations. The $EC_{50}$ values were calculated from the titration series using a modified hill equation,

$$y = \min. y + \frac{(\max. y - \min. y) \cdot x^n}{k^n + x^n} \tag{1}$$

where $n$ is the hill coefficient, in our case $n = 1$ as the ligand to receptor stoichiometry is 1:1. $k$ is the $EC_{50}$ value.

**Cell culture, transfection and labeling.** Chinese hamster ovary cell line (CHO-K1 cells) were cultured using DMEM/F12 medium (PAN Biotech, Aidenbach, Germany/Thermo Fischer Scientific, Dreieich, Germany) and human embryonic kidney cells (HEK-293T) were cultured using DMEM medium (Life Technologies, Darmstadt, Germany/Thermo Fischer Scientific, Dreieich, Germany) both supplemented with 10% (vol/vol) Fetal Calf Serum (Biochrom, Berlin, Germany), Penicillin and Streptomycin (Thermo Fischer Scientific, Dreieich, Germany) and L-glutamine (PAN Biotech, Aidenbach, Germany) at 37 °C and 5% (vol/vol) $CO_2$. Cells were seeded at a density of $2 \times 10^5$ per well on 24 mm circular coverglass (# 1.5A. Hartenstein, Würzburg, Germany) or $2.5 \times 10^3$ on four-well chambered coverglass (Nunc LabTek 1.5 chambered coverglass Cat. No. 155383, Thermo Fischer Scientific, Dreieich, Germany/ CellView 1.5 chambered coverglass C4-1.5-H-N, IBL Labor, Gerasdorf bei Wien, Austria) and grown overnight. The circular coverglasses were cleaned to minimize background fluorescence by sonicating them with 5 M NaOH (Roth, Karlsruhe, Germany) and subsequently by chloroform (PanReac AppliChem, Darmstadt, Germany) for 1 h each followed by drying and storage in absolute ethanol (Merck, Darmstadt, Germany). In the case of HEK-293T cells, the circular/chambered coverglasses were further treated with Poly-D-Lysine (53 µM with incubation for 10 min followed by washing once with PBS (Life Technologies, Darmstadt, Germany) and air-drying). For meausurements, the coverslips were placed in an attofluor cell chamber (Thermo Fischer Scientific, Dreieich, Germany) with culture media containing 15 mM HEPES (Life Technologies, Darmstadt, Germany) without phenol red. For spectroscopy experiments, all constructs except TAG were transfected into CHO-K1 cells whereas TAG was transfected into HEK-293T cells. Transfection in CHO-K1 cells was achieved using Lipofectamine 2000 reagent (Life Technologies, Darmstadt, Germany) in accordance with the manufacturer´s protocol. 2 µg plasmid DNA and 6 µg Lipofectamine 2000 (Thermo Fischer Scientific, Dreieich, Germany) were used per coverslip. In HEK-293T cells, transfection was performed using jetPRIME (Polyplus-transfection SA, Illkirch, France) according to the manufacturer's protocol. TAG was cotransfected with equal amounts of construct DNA and the tRNA plasmid (pCMV tRNA$^{Pyl}$/NESPylRS$^{AF}$). When seeded in 4-chambered coverglass the reagents were scaled accordingly. Right before measurement, in the case of S, cells were labeled with 1 µM AF488 (SNAP-Surface Alexa Fluor 488, New England Biolabs, MA, USA) for 30 min followed by washing thrice with PBS and in the case of TAG, cells were labeled with ~400 nM ATTO488 (6-Methyl-Tetrazine-ATTO-488 (Me-Tet-ATTO488), Jena Bioscience, Jena, Germany) for 10 min followed by washing thrice with PBS. Measurement was performed 8–12 h post transfection.

**Western blot.** For Western blots, HEK293T cells or CHO-K1 cells were seeded in a 15 cm diameter dish were transfected transiently with NT or S with Lipofectamine 2000 (Life Technologies, Darmstadt), according to the manufacturer's instructions. In the case of CHO-K1 cells, double the amount of vector DNA was used for transfection. 48 h post transfection, cells were washed twice with PBS and placed on ice. After addition of 1 mL of lysis buffer (250 µM TRIS, 100 µM EDTA, 100 µM PMSF, 20 µg/mL trypsin, and 60 µg/mL benzamidine), cells were scraped off the plate and the suspension collected. The cells were lysed for 20 s by sonication (Bandelin Sonopuls HD200) and 100 µL were removed (whole lysate sample, W). Separation of soluble, cytosolic cell components from the insoluble membrane debris was done via ultracentrifugation for 20 min at 157,000×g in a Beckman TLA 120.2 (Beckman Optima TLX) at 4 °C. The supernatant (cytosolic fraction, C) was removed carefully and the pellet (membrane fraction, M) was solved in 200 µL of lysis buffer. 10 µL of each sample and the controls were separated on two 12.5% SDS gels (one for the anti-GFP and one for the anti-$\beta_2$AR blot) and transferred onto a PVDF membrane (0.22 µm, Sartorius). The membrane was blocked for 1.5 h at room temperature in 100 mL of TBST buffer (100 mM NaCl, 0.01% Tween 20, 30 mM Tris pH 7.6) supplemented with 5% blocking solution (100 mM NaCl, 50 g/L dry milk powder, 0.01% Tween 20, 30 mM Tris pH 7.6). The buffer was removed and the membrane washed 3× for 10 min each with TBST buffer. Next, the primary antibodies were added and incubated overnight at 4 °C: anti-GFP (Abcam, order no. ab32146) was diluted 1:5000 in TBST supplemented with 5 % BSA; anti-SNAP (NEB, order no. P9310S) was diluted 1:1000 in TBST

supplemented with 5% BSA; anti-$\beta_2$AR (Abcam, order no. ab61778) was diluted 1:2000 in TBST supplemented with 5% BSA. After washing 3× for 10 min with TBST, the secondary antibody HRP Goat Anti-Rabbit (Abcam, order no. ab205718) was diluted 1:2000 in TBST supplemented with 5% BSA and incubated for 1.5 h at room temperature, followed by 3× washing for 10 min each with TBST. 2 mL of ECL Western Blotting Substrate (Pierce, Thermo Scientific) was added and the image captured using the Vilber Fusion FX (Vilber Lourmat, Collégien, France). For G$\beta$ staining, the PVDF membrane post imaging was stripped from its antibodies by incubating with stripping buffer (100 mM glycine, 3.5 mM SDS pH 2.5) for 1.5 h at room temperature. This was followed by 3× washing for 2 min each with TBST and then incubated overnight in 100 mL of TBST buffer (100 mM NaCl, 0.01% Tween 20, 30 mM Tris pH 7.6) supplemented with 5% blocking solution (100 mM NaCl, 50 g/L dry milk powder, 0.01% Tween 20, 30 mM Tris pH 7.6). The buffer was removed and the membrane washed 3× for 10 min with TBST buffer and the primary antibody was added at incubated for 1.5 h at room temperature: anti-G$\beta$ (sc-166123, Santa Cruz Biotechnology, Inc., Dallas, USA) was diluted 1:5000 in TBST supplemented with 5% BSA. The antibody solution was removed and the blots were washed 3× for 10 min each with TBST buffer. Then the secondary antibody was added and incubated for 1.5 h at room temperature, dilution: 1:10,000 in TBST supplemented with 5% BSA., This was followed by 3× washing for 10 min each with TBST imaging as mentioned above using ECL substrate.

**Confocal imaging**. Confocal imaging of transfected cells was performed on a laser scanning confocal microscope (TCS-SP8, Leica Microsystems, Mannheim, Germany) inverted confocal microscope equipped with an HC PL APO CS2 ×63/1.40 oil objective. The sample was excited at 488 nm using an Ar$^+$ laser (5 μW at the back focal plane). Images were scanned bidirectionally at 100 Hz (pixel size of 43 nm × 43 nm in accordance with the Nyquist criterion) and a pinhole of 1 AU.

**Time-resolved setup**. A custom-built confocal setup based on an Olympus IX 71 stand (Olympus, Hamburg, Germany) equipped with a Time-Correlated Single Photon Counting (TCSPC) system (Hydraharp 400, Picoquant, Berlin, Germany) were used to perform FCS and fluorescence anisotropy measurements. Briefly, the excitation laser (485 nm pulsed laser LDH-D-C-485, Picoquant) was fiber coupled (Laser Combining Unit with polarization maintaining single mode fiber, Pico-Quant, Berlin, Germany) and expanded to a diameter of 5.5 mm by a telescope to fill the back aperture of the objective (×100 oil immersion, NA 1.49, UAPON100xOTIRF, Olympus, Hamburg, Germany) and thus create a diffraction limited focal spot. The size of the effective volume element is 0.5 femtolitre. Before entering the objective lens the laser polarization was adjusted by an achromatic half-wave plate (AHWP05M-600, Thorlabs, Bergkirchen, Germany) and depolarization in the excitation path was minimized by a polarizing beamsplitter (PBS101 420-680, Thorlabs, Bergkirchen, Germany). A beamsplitter (quad band zt405/473-488/561/640 rpc phase r uf1, AHF, Tübingen, Germany) guides the laser through the objective epi-illuminating the sample. In the detection path a 50 μm/100 μm pinhole (PNH-50/PNH-100, Newport, Darmstadt, Germany) rejected out of focus light before projected on photon counting detectors (2× PMA Hybrid-40, Picoquant, Berlin, Germany) by a telescope in a $4f$ configuration (focal length of lenses: 60 mm, G063126000, Qioptiq, Rhyl, UK). The beam was split via a polarizing beamsplitter cube (10FC16PB.3, Newport, Darmstadt, Germany) in parallel (detector 1) and perpendicular emission (detector 2) after the first lens of the telescope. Dichroic beamsplitters (Beamsplitter T 635 LPXR, AHF, Tübingen, Germany) and Emission filters (band pass filter Brightline HC 525/50 AHF, Tübingen, Germany) reject unspecific light in each detection path.

**Time-resolved data acquisition**. The laser spot was focused on the membrane of the cells. Unless stated otherwise, measurements were carried out on the basolateral membrane. All data were acquired using SymPhoTime 64 (PicoQuant, Berlin, Germany). For FCS and TRA measurements the laser was operated in pulsed mode at 20 MHz and for fullFCS the laser was operated in continuous wave mode. In all cases the sample binning was set to 4 ps. FCS and TRA data were acquired for 5–10 min depending on photon count rate. fullFCS data in the case of NT was acquired for 20 min and in the case of S was acquired for 40 min to benefit from the high signal to noise ratio of the organic fluorophore.

**Theoretical concept and calibration**. The effective volume ($V_{eff}$) in a confocal setup is

$$V_{eff} = \pi^{3/2} \cdot z_0 \cdot \omega_0^2 \tag{2}$$

where $z_0$ is the axial width of the effective confocal volume, $\omega_0$ the lateral width of the effective confocal volume[66].

$V_{eff}$ was determined as described before[66] by diffusion analysis using AlexaFluor 488 (AF488; 2 nM) in ddH$_2$O by fitting with

$$G(\tau) = G_0 \left[ 1 - T + T \cdot \exp\left(-\frac{\tau}{\tau_T}\right) \right] \cdot \left( \left(\frac{1}{1+\tau/\tau_D}\right)\left(\frac{1}{\sqrt{1+z_0^2/\omega_0^2(\tau/\tau_D)}}\right) \right) \tag{3}$$

where $G_0$ is the correlation amplitude, $\tau$ the lag time, $T$ the fractions of molecules in the triplet state (dark state), $\tau_T$ the lifetime of the $T$, and $\tau_D$ the diffusion time.

Correction factors were determined to account for fluorescence depolarization due to high-NA objective based on ref. [67]. In our setup, they were $l_1 = 0.3329$, $l_2 = 0.1613$ for 485 nm excitation.

**Data analysis**

*FCS*. The single photon traces acquired were autocorrelated and exported to text files using SymPhoTime 64. Correlation curve fits were performed using Origin Pro (OriginLab, MA, USA) using three different fit models with s.e.m as the weighting factor:

 i. 2D3DT: one 2D diffusion component, one 3D diffusion component, one Triplet (T, exponential) component T

 ii. 22DT: two 2D diffusion components, one T component

 iii. 2D2T: one 2D diffusion component, one T and one additional exponential component.

The general equation is

$$G(\tau) = \left(\frac{1}{N}\right)\left( a\left(\frac{x_1}{1+\tau/\tau_1}\right) + b\left(\frac{1-x_1}{1+\tau/\tau_2}\right) + c\left(\frac{1-x_1}{1+\tau/\tau_2}\right)\left(\frac{1}{\sqrt{1+\omega_0^2/z_0^2(\tau/\tau_2)}}\right) \right)$$
$$\prod_{i=1}^{n}\left(1 - T_i + T_i \cdot \exp\left(-\frac{\tau}{\tau_{Ti}}\right)\right) + G_\infty \tag{4}$$

where $N$ is the number of molecules in the focus volume, $\tau$ is the lag time or delay time, $T_i$ is the dark triplet state fractions or protonated fractions of molecules, $\tau_{Ti}$ is the lifetime of singlet/triplet or protonated/deprotonated transition dynamics, $x_1$ is the fraction of the 1st diffusion component, $\tau_1$ is the corresponding diffusion time, $1 - x_1$ is the fraction of the 2nd diffusion component, $\tau_2$ is its corresponding diffusion time, $z_0$ is the axial width of the effective confocal volume, $\omega_0$ is the lateral width of the effective confocal volume and $G_\infty$ is the offset. $a$, $b$ and $c$ are constants that differ for each model. Their values are summarized in Table 1.

With the extracted diffusion times from the fit, and $\omega_0$ from the calibration, the diffusion coefficient $D$ can be calculated using the equation

$$D = \frac{\omega_0^2}{4 \cdot \tau_D} \tag{5}$$

In some cases, $D$ was weighted over the corresponding molecules exhibiting it as a way to normalize using the following equation:

$$D_{AVG,i} = D_i \cdot x_i \tag{6}$$

where $i$ was the respective component.

*Time-resolved fluorescence anisotropy*. The single photon arrival times acquired were recorded and exported to text files using SymPhoTime 64. Decay curve fits were performed using MATLAB (Mathworks, MA, USA) using custom scripts and fitted globally as previously described[43]:

$$I_{VV}(t) = \frac{1}{3}I_0\left[\sum_{i=1}^{n} F_{fl,i} \cdot \exp\left(-\frac{t}{\tau_{fl,i}}\right)\right]\left[1 + 2 \cdot \sum_{j=1}^{m} r_{int,j} \cdot \exp\left(-\frac{t}{\tau_{rot,j}}\right)\right] + BG_{VV} \tag{7}$$

$$g \cdot I_{VH}(t) = \frac{1}{3}I_0\left[\sum_{i=1}^{n} F_{fl,i} \cdot \exp\left(-\frac{t}{\tau_{fl,i}}\right)\right]\left[1 - \sum_{j=1}^{m} r_{int,j} \cdot \exp\left(-\frac{t}{\tau_{rot,j}}\right)\right] + BG_{VH} \tag{8}$$

$I_0$ is the overall emission amplitude, $F_{fl,i}$ and $\tau_{fl,i}$ are the fraction and fluorescence lifetime attributed to excited state population decay component $i$, $r_{int,j}$ is the initial anisotropy (here fixed to the fundamental anisotropy $r_{int,j} = 0.38$) and $\tau_{rot,j}$ is the rotational relaxation time of the component $j$, respectively. The background $BG$ is the only parameter that was fit independently for both channels, all other were shared between the channels. $g$ is the ratio of channel sensitivities (g-factor) and was calibrated before each measurement.

*Full fluorescence correlation spectroscopy*. In the case of fullFCS, the intensity traces were split into pieces (5–60 s range) and each piece was correlated based on ref. [47]

**Table 1 Values of constants in Eq. (4) for different FCS fit models.**

| Model | a | b | c | n |
|---|---|---|---|---|
| i | 1 | 0 | 1 | 1 |
| ii | 1 | 1 | 0 | 1 |
| iii | 1 | 0 | 0 | 2 |

with customizations to take membrane protein dynamics into consideration. The custom scripts can be found at https://github.com/khemmen/katcorr/; https://doi.org/10.5281/zenodo.5786498. In our case, instead of considering all the curves for averaging ($G_{j\neq k}$ in ref. [47]) we use only the first $x$ curves (depending upon the split size, $x = 5$–$30$). This is based on our assumption that in the case of membrane proteins, due to their decreased mobility[9], the first minutes reflect the ground truth before photobleaching, or moving of the cell, comes into play. In turn the mean square deviation $d$ in our case would equate to the following:

$$d = \frac{(A_n - A_{\text{avg}})^2}{n\_\text{points}} \tag{9}$$

where $d$ is normalized to the amounts of data points ($n\_\text{points}$) used to calculate $A_n$, $A_n$ is the individual average of each of the $n$ curves and $A_{\text{avg}}$ is the average of the first $x$ curves. The value of $x$ was adjusted such that the macro time between comparisons were the same, e.g. for the 10 s slices 30 curves were averaged and for the 60 s slices the first 5 curves were averaged.

The obtained correlation curves for each measurement were globally fitted with the same model as for the pulsed FCS experiments (Eq. (4)) except for the crosscorrelation function where we added the additionally required photon antibunching term:

$$G_{\text{CCF}}(\tau) = \left(\frac{1}{N}\right) \cdot \left(\left(\frac{x_1}{1+\tau/\tau_1}\right)\left(\frac{1-x_1}{1+\tau/\tau_2}\right)\left(\frac{1}{\sqrt{1+1/s^2(\tau/\tau_2)}}\right)\right) \cdot \left(-a_{\text{ab}}\exp\left(-\frac{\tau}{\tau_{\text{ab}}}\right)\right) \cdot$$
$$\left(1 - \sum_{i=1}^{n}\left[a_{\text{R}i} - a_{bi}\exp\left(-\frac{\tau}{\tau_{\text{R}i}}\right)\right]\right) + G_{\infty} \tag{10}$$

where $a_{\text{ab}}$ denotes the amplitude of the photon bunching (usually ~1) and $\tau_{ab}$ is related to the fluorescence lifetime. Note that here usually several relaxation times $\tau_{\text{R}i}$ (3 and 4 in our case for EGFP and SNAP constructs, respectively) are required as for pulsed FCS due to the extended measurement range.

In the global fit, the relaxation times for all curves corresponding to each measurement were jointly fitted while $\tau_1$, $\tau_2$, $N$, and $x_1$ were shared among each time slice. Only the amplitudes $a_{\text{R}i}$ of $\tau_{\text{R}i}$ were individually optimized. This global, joint fitting approach reduced the number of fitting parameters drastically and stabilized the results.

Supplementary Fig. 14 shows schematically the data selection and fitting procedure.

To compare the absolute, polarization dependent differences between $\text{ACF}_{\text{VV}}$ and $\text{ACF}_{\text{VH}}$, $\triangle\text{ACF}(\tau)$ we extracted the relaxation kinetics $A_{\text{R}}(\tau)$ for both ACFs:

$$A_{\text{R}}(\tau) = 1 - \sum_{i=1}^{n}\left[a_{\text{R}i} - a_{bi}\exp\left(-\frac{\tau}{\tau_{\text{R}i}}\right)\right] \tag{11}$$

$$\triangle\text{ACF}(\tau) = |A_{\text{R,VV}}(\tau) - A_{\text{R,VH}}(\tau)| \tag{12}$$

*Statistics and reproducibility.* Origin Pro (OriginLab, MA, USA) was used for statistical analysis. Pairwise comparisons were performed using Two-sample *t*-test with a null hypothesis that the means of the distribution were equal. The box plots used in the figures represent a box with a size of 25–75% of the plotted dataset and the whiskers are within the 1.5 interquartile range for the given dataset. The specific data points are plotted over the box plots. The live-cell experiments were performed with a sample size of up to 10 cells per measurement condition and the data was acquired on different days to account for reproducibility.

**Reporting summary**. Further information on research design is available in the Nature Research Reporting Summary linked to this article.

## Data availability

All datasets plotted in the main figures are available in Supplementary Data 1 and the related raw acquisition data is available upon request from the corresponding authors. The unprocessed western blot images corresponding to Fig. 2d, Supplementary Figs. 10 and 11 are given in Supplementary Figs. 21 and 22, respectively.

## Code availability

The custom scripts used in the study can be found at https://github.com/khemmen/katcorr/ and at Zenodo[68] with https://doi.org/10.5281/zenodo.5786498.

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

## Acknowledgements

We thank Edward Lemke and Gemma Estrada Girona (European Molecular Biology Laboratory) for the gift of the pCMV tRNA^Pyl/NESPylRS^AF plasmid and expert training on how to use it. This project was funded by the DFG ReceptorLight TRR166 (project C06 to M.L., K.G.H., and A.B.). We thank Ulrike Zabel (Institute of Pharmacology, University of Würzburg) for her help in designing the EGFP tagged β₂-AR plasmids. We thank Julia Wagner for help with the apical and basal FCS measurements and Thomas-Otavio Peulen (University of California, San Francisco) for help with the tttrlib-library used for scripting the fullFCS analysis. We thank Thorsten Wohland (National University of Singapore) for his fruitful discussions. G.B. and M.S. acknowledge funding by the Deutsche Forschungsgemeinschaft (DFG, project SA829/19-1). We thank the Core Unit Fluorescence Imaging, University of Wuerzburg for giving us access to the Leica TCS-SP8. This publication was supported by the Open Access Publication Fund of the University of Wuerzburg.

## Author contributions

K.G.H. and M.J.L. designed research; A.B., S.C., J.H.K. performed time-resolved experiments; A.B. and K.H. analyzed the data, A.B. and K.J. performed the Western Blotting, M.F. built and calibrated the experimental spectroscopic setup, G.B. and M.S. designed the unnatural amino-acid constructs; K.G.H., K.H., A.B. and M.J.L. wrote the manuscript.

## Funding

## Competing interests

The authors declare no competing interests.
