## [Peer Review File · Communications Biology]

Reviewers' comments:

Reviewer #1 (Remarks to the Author):

The authors provide a well-constructed and experimentally well-conducted manuscript focusing on understanding the mobility of GPCRs, specifically the beta2-adrenergic receptor and to a lesser extent the alpha2-AR. Using a rig they built themselves, they could tag the receptor in different ways and examine receptor rotational and diffusional mobility by time resolved anisotropy, fluorescence correlation microscopy and full continuous wave FCS. Because they are using a single instrument that can capture events from the psec to second range. This is impressive. Further, they show to a limited extent that the tagged receptors remain functional. However, I am not certain what if any biological questions the work answers. They suggest that some of the events are correlated with biological events like cellular signalling or receptor internalization- but these remain correlations and it is not clear to me at all what new questions the increased temporal resolution is poised to answer- especially since they are using heterologous expression systems. I suppose this is my most important criticism. Other minor issues include:

- 1) Concentration-response data for the cAMP measures demonstrating functionality and a similar experiment for the alpha2AR. Even if the primary signalling pathway is not compromised how does this help us understand events like biased signalling or receptor dimerization?
- 2) Change the legend to Figure 1b- the effects of these ligands are not shown here, only their structures.
- 3) On page 7- the title of the last section "Ligand stimulation of beta2-AR affects diffusion constants" contradicts directly the title of Figure 5. Which is it?
- 4) The authors discuss dimerization but it is not clear which experiments examine it directly.

Reviewer #3 (Remarks to the Author):

Review of Comm Bio 8871

"Unraveling the hidden temporal range of fast b2-adrenergic receptor mobility by time-resolved fluorescence"

Summary:

•This paper characterizes the spatial and temporal dynamics of the b2AR in the plasma membrane using tagged b2ARs and several spectroscopic techniques.

The aim of the study is to characterize the movement of the b2AR in the plasma membrane. The study was conducted using GFP tagged, SNAPtagged or biorthogonal labelled b2AR, with tags in different positions. Movement of the b2AR was assessed using fluorescence correlation spectroscopy, time-resolved anisotropy and polarised fluorescent correlation spectroscopy to determine the receptor mobility.

The tags were assessed by cAMP assay, and suggested not to impact the normal functioning of the tagged b2AR relative to WT b2AR.

FCS showed that the receptors had a fast and slow lateral movement components.

FP showed that the receptors had a long and short rotational movement. The long was associated with movement of the tagged-b2AR and the short was associated with movement of the tag alone. The number of receptors in the membrane did not influence the mobility of the receptors.

The NT receptor showed reduced diffusion with ISO but not SAL or CAR compared to unstimulated NT-b2AR. This result was not observed when S was used as the tagged receptor.

The conclusions of the study are that different pools of b2AR exists within plasma and intracellular membranes that have different mobility properties.

Overall Impression of the manuscript:

This manuscript explores the mobility of the b2AR expressed in CHO-K1 or HEK293T cells. This is quite a complicated study, that builds upon previous studies using alternative techniques. It identifies a previously unappreciated fast rotational motion of the b2AR. It increases our understanding about how the b2AR transitions about the plasma membrane. Most of the work is sound, however, there are minor inconsistencies with different cell backgrounds used for different assays (ie. CHO-K1 for spectroscopy v HEK293T for Western blots), which may influence the results.

Specific comments:

1. Figure S1 was difficult to understand and I found the figure legend confusing. Is b) the cAMP response in CHO cells? Is d) the cAMP response in HEK293T cells? Could you please add titles to the graphs so it is easier to see what results are from each cell line.
2. I could not find the methods for the cAMP fluorometric assay in the methods section. Could you please check that this is in the methods section. And if it is not, could you please add it to the methods section?
3. Why are there 2 standard curves for the cAMP assay (Figure S1a and S1c)?
4. Why do the two standard curves have different Emax (200cnts vs 40 cnts)?
5. Compared to wild type in Fig S1b, it looks like the IL3 tag reduces the basal constitutive activity of the b2AR compared to the WT (80 cnts for IL3 vs 55 cnts for WT), which is a difference. Also the net cAMP response looks different by eye for the IL3 construct compared to the WT (80-15 = 65 cnts for IL3 vs 55-10 = 45 cnts for WT). Could the authors please explain why they think that there are no functional differences between the differently tagged receptors, when this appears to be a functional difference?
6. I can't find the statistics to support the conclusion by the authors that the cAMP responses are unaltered. Could you please do the statistical tests on the data presented in Figure S1?
7. Table S1 shows the interpolated cAMP concentrations. The cAMP response in the presence of isoproterenol at the IL3 (15 cnts), S (10 cnts) and WT (12 cnts) in figure S1b seem to be outside the linear range of the cAMP standard curve shown in figure S1a, therefore they may be unreliable. Could the authors please check that these values are indeed within linearity? If not, the assays would need to be repeated to ensure all data points are within the linear range of the assay.
8. For Figure 2/Figure S8 why were the Western blots done with HEK293T cells, when the cAMP and diffusion experiments for the NT b2AR were done in a CHO cell background? These should have been done in the same cell background, since the cell background could influence the trafficking/mobility of the receptors. Could you please show the same Western blot data for the receptors expressed in the CHO cell background.
9. Since the S receptor appears to be behaving differently, did the authors do the Western blots for the S receptor? How did this receptor compartmentalize compared to the NT receptor?
10. Can the S receptor internalize with the organic fluorophore bound? And in particular does it internalize when stimulated with ISO?

Dear Reviewers,

We thank you for your appreciation and constructive critiques on our work. We have carefully addressed all the questions posed and have presented new experiments where relevant. Please find all answers in green and the respective text changes in the main manuscript indicated in blue.

Reviewer #1's comment:

The authors provide a well-constructed and experimentally well-conducted manuscript focusing on understanding the mobility of GPCRs, specifically the beta2-adrenergic receptor and to a lesser extent the alpha2-AR. Using a rig they built themselves, they could tag the receptor in different ways and examine receptor rotational and diffusional mobility by time resolved anisotropy, fluorescence correlation microscopy and full continuous wave FCS. Because they are using a single instrument that can capture events from the psec to second range. This is impressive. Further, they show to a limited extent that the tagged receptors remain functional. However, I am not certain what if any biological questions the work answers. They suggest that some of the events are correlated with biological events like cellular signalling or receptor internalization- but these remain correlations and it is not clear to me at all what new questions the increased temporal resolution is poised to answer- especially since they are using heterologous expression systems. I suppose this is my most important criticism.

We thank the reviewer for the appreciation and comments on our work.

To observe receptor mobility, it is necessary to label the receptor. This can be achieved either via modifications of the receptor itself (i.e. by conjugating it to a fluorescent protein or a tag which can later be bound to a fluorophore) and transfection of the modified construct, or by labeling them e.g. with fluorescent ligands or antibodies. While the former approach, which is the one we chose, has the drawback that it necessitates transfection, it has the advantage of essentially 100% labeling efficiency and of constant labeling. The latter approaches, in contrast, suffer from incomplete and generally only reversible labeling; in addition, ligands and antibodies have the potential of modifying the activation state of a receptor and thereby modify its behavior, including mobility. Since we aimed to investigate the effects of ligands on mobility, this precluded the use of labeled ligands as a means of observing the receptors.

However, in order to mitigate the drawbacks of using heterologous expression, we now additionally transfected the NT construct in HEK293T cells (Figure R1), which showed comparable mobility. The data shown here (see Figure R1 below) is also included now in the revised manuscript as **fig. S6**, table S2, table S3 and in the highlighted part in page 4 of the main manuscript.

The reason why we originally chose CHO-K1 cells is due to their robustness in handling and measurement; in addition, they have no background of endogenous adrenergic receptors, which avoids the complication of non-observed background receptors that might interfere with our measurements. Thus, the shown spectroscopy measurements on CHO-K1 cells are representative here.

Figure R1. Diffusion parameters of HEK293T and CHO-K1 cells expressing NT-EGFP. (a) Both cell lines exhibit a fast and slow translational diffusion constant. Translational diffusion constant of plasma membrane bound receptors from CHO-K1 cells have a narrower distribution over HEK293T cells. HEK293T cells exhibit, $D_{fast} = 8.35 \pm 5.34 \mu\text{m}^2\text{s}^{-1}$, $D_{slow} = 0.25 \pm 0.18 \mu\text{m}^2\text{s}^{-1}$ for $n = 10$; CHO-K1 cells exhibit, $D_{fast} = 6.08 \pm 2.80 \mu\text{m}^2\text{s}^{-1}$, $D_{slow} = 0.10 \pm 0.02 \mu\text{m}^2\text{s}^{-1}$ for $n = 12$; (b) The fraction of the fast diffusion constant is not different in both cell lines. X_{fast} for HEK293T cells is 0.36 ± 0.07 and for CHO-K1 cells is 0.33 ± 0.05 (c) The rotational correlation times of receptors in HEK293T cells have a narrower distribution compared to CHO-K1 cells. HEK293T cells show $\tau_{rot} = 58 \pm 54$ ns while CHO-K1 cells show $\tau_{rot} = 114 \pm 71$ ns.

The reviewer also asks about the biological relevance of the findings. We have, therefore, expanded the discussion section to cover interpretations of our data more extensively. As pointed out, fundamental questions about cellular signalling or receptor internalization strongly depend on the underlying rate constants. Here, we devised technologies to highlight temporal dynamics of a model membrane receptor in the cell membrane. We believe that such measurements are a necessary foundation for upcoming fluorescence based inter-/ intra-molecular studies such as receptors meeting and interacting with their downstream signalling partners (G-proteins, GRKs and β -arrestins) and when, how and how long they may interact with other cell surface proteins including homologous and heterologous receptors.

Other minor issues include:

1) Concentration-response data for the cAMP measures demonstrating functionality and a similar experiment for the alpha2AR. Even if the primary signalling pathway is not compromised how does this help us understand events like biased signalling or receptor dimerization?

This is a valid concern. To demonstrate the unaltered functionality of our used constructs, we repeated the cAMP measurement as a concentration series with triplicates from independent measurements to show that the constructs used in this study do indeed exhibit a similar EC_{50} value for activation by agonist to that of the wild type. Figure R2 shows that the constructs used in this study do indeed exhibit similar EC_{50} values in the range of 1 – 100 nM when activating with the agonist isoproterenol. These EC values are comparable to the WT ($EC_{50} = 1.3$ nM), and similar to the values reported earlier ¹. **Fig. S1** and **table S1** in the supplementary are revised accordingly and show the cAMP response to a concentration series of isoproterenol. The materials and methods section has been updated describing in more details the fluorometric cAMP measurements (page 11 para 5).

The reviewer raises the important question on how the assurance of non-compromised primary signalling pathways relates to the understanding of biased signalling or receptor dimerization. On the first glance these aspects are unrelated; however, as commonly accepted, we used the cAMP assay

here primarily to test whether the introduction of the fluorescent labels hampers the primary / basic function of the receptor. This was not the case here because of a similar cAMP response of the different fluorescent constructs to the WT. The receptor diffusion and possible dimer-/oligomerization was then subsequently probed by our fluorescence spectroscopic approaches, FCS and TRA. While N- and C-terminal labels are well tolerated with respect to signalling, as has been reported previously, there is a reduction of potency of the IL3-construct by about 1 log unit; however, it should be noted that full activation is also achieved with this construct, illustrating that maximal cAMP levels were achieved as with the wild type construct, illustrating that despite the labelling in a region critical for G-protein-coupling, also this construct has a significant degree of functionality.

Figure R2. cAMP response was measured using the commercial Abcam fluorimetric cAMP assay (Abcam, ab138880) with activation using different concentrations of ligand (Isoproterenol) ranging from 100 pM to 10 μ M with a factor 10 step size. cAMP concentration response for different ranges of Isoproterenol for. **(a)** WT and NT **(b)** WT and IL3 **(c)** WT and TAG **(d)** WT_A and NT_A. WT data shown in (a) and (b) are the same. All cases except (c) were expressed in CHO-K1 cells. Both WT and TAG in (c) were expressed in HEK 293T cells. The data were measured as triplicates.

Construct	EC50 (nM)
WT	1.3 \pm 0.8
NT	4.9 \pm 3.7
IL3	32.9 \pm 19.5
WT	15.3 \pm 3.8
TAG	2.1 \pm 0.0
WT_A	88.0 \pm 66.3
NT_A	39.1 \pm 37.5

2) Change the legend to Figure 1b- the effects of these ligands are not shown here, only their structures.

Done. We thank the reviewer for pointing this out, and have now corrected the figure legend (page 22).

3) On page 7- the title of the last section "Ligand stimulation of beta2-AR affects diffusion constants" contradicts directly the title of Figure 5. Which is it?

We apologize for this mistake. Our study shows that ligand stimulation indeed affects diffusion constants. The title of Figure 5 has been changed accordingly in the revised manuscript (page 26).

4) The authors discuss dimerization but it is not clear which experiments examine it directly.

We apologize for not being clear on this topic. We analyzed and discussed the steady state anisotropy data which is an accepted measure for probing changes in oligomerization^{2,3}. On the one hand small fluorophores like AF488 and ATTO488 increase their steady state anisotropy when bound to larger proteins; on the other hand, steady state anisotropy generally decreases when multiple fluorophores are in closest vicinity (1-10 nm) e.g. when forming an oligomer (homoFRET). Thus, a low steady state anisotropy when monitoring fluorescently labelled receptor complexes would suggest a significant degree of oligomerization.

In our work, we see a low steady state anisotropy of 0.1 (and below) for basal and activated states. Thus we assume an inherent oligomerization in the basal state that does not vary upon activation. The text has been rephrased accordingly in the revised manuscript (page 6 para 3, page 8 para 3, page 11 para 2).

Reviewer #2's comments:

The manuscript is based on Fluorescence Fluctuation Spectroscopy assays to analyze β_2 adrenergic receptor dynamics. The authors confirmed previously reported diffusion constants, but they also propose a reinterpretation for rotational diffusion data that could fit β_2 adrenergic receptor dynamics to the classic Saffman-Delbrück model for mobility of membrane proteins. The authors also highlight a high basal intracellular receptor mobility originated from transport vesicles. Effect of ligands with different pharmacological properties was also analyzed. The manuscript shows the importance of the labeling strategy on receptor mobility and brings advances on the usage of combining different fluorescent spectroscopy techniques. Data shown in the manuscript add to the understanding of GPCR non-canonical signaling at subcellular level. I have minor issues to be addressed:

We would like to thank the reviewer for the kind appreciation of our work.

1) In the Abstract and Conclusion authors claim that the temporal range for Beta2-adrenergic receptor mobility (nanosecond to second) is unusual for a cellular protein. What about membrane proteins that are also trafficked from ER through transport vesicles? RTKs are membrane receptors that are also internalized through clathrin-coated pits. Should not such parameters be expected to be similar between GPCRs and RTKs?

We apologize for phrasing our abstract in such a way that it conveyed that β_2 -AR acts different from other cell surface (and trafficked) proteins in exhibiting mobility from ns to s and that is unique to it. In fact, such mobility would be expected from any membrane protein including RTKs, even though structural differences (single vs. multi-transmembrane helices, mono-, vs. di- vs. oligomerization etc) may modify the overall behavior. First, we would like to make the point that the fluorescence technique and assay need to be smartly chosen so that they do not hamper the temporal resolution required in the experiment. Second, when probing membrane proteins we expect to see at least two different

mobility constants originating from membrane diffusion and vesicle transport. The abstract and conclusion have been updated in the revised manuscript accordingly (page 2, page 11 para 3).

2) Authors claim that “One would expect an increase in the fraction of D_{fast} due to internalization, but this is counteracted by the clathrin coated pit formation which leads to receptors clogging up on the plasma membrane. This would in turn lead to a decrease in D_{slow} as shown by NT upon agonist activation, which gives us access to all receptors relative to S”. Have authors tested internalization inhibition (e.g. DynK44A) or potentialization (e.g. constitutively active Rab5) effect on D_{slow} and D_{fast} ?

We have thought about this important issue carefully and set up additional experiments to show the effect of internalization, see Figure R3 and R4 below and **fig. S20** in the revised manuscript (also text in page 8, para 1). In order to inhibit internalization the well-known inhibitor pitstop2⁴ was used. As expected we observe a significant decrease in the slow diffusion constant of NT similar to treatment with ISO. Pitstop2 is known to prevent both clathrin dependent⁵ and clathrin independent endocytosis⁶ so that the receptors on the plasma membrane clog up over time leading to the observed decrease.

Figure R3. Diffusion parameters of CHO-K1 cells expressing NT-EGFP before (green) and after (green with bricked background) being treated with pitstop2, an inhibitor of internalization. The cells were treated with a final concentration of 25 μM pitstop2 and incubated for 30 min at 37 $^{\circ}\text{C}$. The measurements were made within the next 30-40 min. (a) The diffusion constant of membrane bound receptors from the untreated to the treated case are different to a significance level of *- $p < 0.0001$. The cells in the untreated case exhibit $D_{slow} = 0.10 \pm 0.02 \mu\text{m}^2\text{s}^{-1}$ for $n = 12$ and the cells in the pitstop2 treated case exhibit $D_{slow} = 0.04 \pm 0.03 \mu\text{m}^2\text{s}^{-1}$ for $n = 12$. The diffusion constant distribution of vesicle bound receptors do not change, however their fraction decreases. (b) The fraction of the fast diffusion constant does not change after treatment with pitstop2. Untreated case shows $x_{fast} = 0.33 \pm 0.05$ and the treated case shows $x_{fast} = 0.32 \pm 0.06$.

Figure R4. Diffusion constant of plasma membrane bound receptors of NT in their untreated state compared to treatment with ISO and pitstop2. ISO increases both clathrin coated pit formation and internalization whereas pitstop2 blocks internalization after clathrin coated pit formation. In both cases, the amount of clustered plasma membrane bound receptors increase over time and this is seen as a decrease in their diffusion constants. Mean is given as yellow box and the line in the box represents the median of the population. Untreated case exhibits $D_{\text{slow}} = 0.10 \pm 0.02 \mu\text{m}^2\text{s}^{-1}$ for $n = 12$, ISO treated case exhibits $D_{\text{slow}} = 0.06 \pm 0.03 \mu\text{m}^2\text{s}^{-1}$ for $n = 15$, pitstop2 treated case exhibits $D_{\text{slow}} = 0.04 \pm 0.03 \mu\text{m}^2\text{s}^{-1}$ for $n = 12$.

Since pitstop2 only stops internalisation at an acute level, the overall equilibrium of internal vesicular vs. cell surface receptors does, unfortunately, not shift drastically in our experimental setting. As a result, the fraction of the fast diffusion constant does not change significantly, although the diffusion constant of the membrane bound receptors slows down by half. Thus, the result of our additional control supports our previous hypothesis but also adds complexity to its interpretation.

About the decrease in D_{slow} upon agonist activation, could this not be attributed to receptor “trapped” at membrane microdomains such as lipid rafts? Could authors comment on these topics?

Although our experiments do not strictly show the presence or influence of membrane micro- or nanodomains articles in literature have addressed the influence of membrane microdomains⁷ and nanodomains⁸⁻¹⁰ on diffusion. In the view of this previous findings, one could speculate that during activation micro or even nanodomains work in tandem with clustering and internalization leading to a decrease in the diffusion constant. The discussion of the revised manuscript has been updated accordingly (page 10 para 3).

3) Authors claim that “The S construct does not show any significant changes, which is in accordance with a recent single particle tracking⁹ and a FRET study⁴⁸. This difference could be attributed to the inherent differences of NT and S”. What would be these “inherent differences”? Fluorophore properties? Authors should mention that.

Thanks for pointing this out. We explain this more thoroughly now in the revised manuscript. We now explain in the main manuscript (page 10 para 2) that the S construct by default has less fluorescent receptors in the membrane relative to NT as only the plasma membrane bound receptors become fluorescent upon labeling. Further incubation steps after washing give time for the normal cellular internalization to play which should eventually decrease the amount of receptors in the plasma membrane which is directly in the diffusion times derived from the FCS curves. It seems that the ratio

of fluorescently clustered receptors to non-clustered receptors in the case of S is eventually less than NT so that the decrease in the slow diffusion constant for S remains masked.

4) It's mentioned that "The rotational diffusion on the other hand seems to get faster upon ligand treatment, at least in the case of NT, as also reported for the FIAsh-tagged α 2A-receptor 23". How can this be explain based on the Pharmacological notion that ligands stabilize the receptor at more restricted conformations? For instance, agonists stabilize active conformations and antagonist stabilize inactive conformations. Could authors discuss on that?

This is an interesting and complex question: there have been numerous and at times contradictory reports on ligand effects on receptor mobility. In addition, as we recently observed in the case of the μ -opioid receptor ⁴, ligand-induced changes in receptor mobility may also be time-dependent and change very rapidly. We believe that they are less related to the ligand-induced changes in conformation, but rather depend on ligand-regulated interactions with other proteins (such as G-proteins, GRKs, β -arrestins and many others), which might also include more or less tight interactions with the actin cytoskeleton ¹¹ and other structural elements of the cell.

We have now covered these considerations more extensively in our discussion section (page 10 para 4, page 11 para 1).

5) Authors claim that there is no dimerization. Beta2-adrenergic receptor dimerization has already been shown (reference 31 cited in the manuscript). Do authors have positive controls for dimerization?

We apologize for not being clear on the topic of dimer-/oligomerization. Our steady state anisotropy data being lower than 0.1 in both the basal and activated states indicate that there is an inherent oligomerization already in the basal state that does not change significantly during activation. Owing to this, we have not performed any positive controls for dimerization. We have rephrased the dimerization/oligomerization part in the revised manuscript to avoid confusion (page 11 para 2).

6) Beta2-adrenergic receptor is a Gs coupled receptor. Would the analyzed parameters be expected to be similar to GPCRs coupled to other G proteins? What about β -arrestin role on the fast diffusion component? Would authors have such complementary results or could they discuss that based on data from literature?

As the reviewer mentioned β ₂-AR is a G_s coupled receptor. Unfortunately, the molecular weight increase for a dimer / or a factor of 2 would not be sufficient to chance the diffusion time significantly. As theorized by the Saffman and Delbrück model ¹² and its approximations ^{13,14}, it is possible to say that β ₂-AR coupled to G_s or β -arrestin would be hard to detect from β ₂-AR and discern between as a function of mobility.

A final consideration in this context is that receptor/G-protein interactions in intact cells are apparently quite short-lived ¹⁵, presumably because high GTP levels in cells lead to dissociation of receptor/G-protein complexes, so that the influence of G-proteins is presumably lower than one would initially expect. These considerations are now also included in the discussion (page 10 para 3).

7) I could not find description of cAMP measurement methodology.

We apologize for this omission and have now added the cAMP measurement methodology in the Materials and Methods section (page 11 para 5).

Reviewer #3's comments:

Summary:

•This paper characterizes the spatial and temporal dynamics of the b2AR in the plasma membrane using tagged b2ARs and several spectroscopic techniques.

The aim of the study is to characterize the movement of the b2AR in the plasma membrane.

The study was conducted using GFP tagged, SNAP tagged or biorthogonal labelled b2AR, with tags in different positions. Movement of the b2AR was assessed using fluorescence correlation spectroscopy, time-resolved anisotropy and polarised fluorescent correlation spectroscopy to determine the receptor mobility.

The tags were assessed by cAMP assay, and suggested not to impact the normal functioning of the tagged b2AR relative to WT b2AR.

FCS showed that the receptors had a fast and slow lateral movement components.

FP showed that the receptors had a long and short rotational movement. The long was associated with movement of the tagged-b2AR and the short was associated with movement of the tag alone. The number of receptors in the membrane did not influence the mobility of the receptors. The NT receptor showed reduced diffusion with ISO but not SAL or CAR compared to unstimulated NT-b2AR. This result was not observed when S was used as the tagged receptor.

The conclusions of the study are that different pools of b2AR exists within plasma and intracellular membranes that have different mobility properties.

Overall Impression of the manuscript:

This manuscript explores the mobility of the b2AR expressed in CHO-K1 or HEK293T cells. This is quite a complicated study, that builds upon previous studies using alternative techniques. It identifies a previously unappreciated fast rotational motion of the b2AR. It increases our understanding about how the b2AR transitions about the plasma membrane. Most of the work is sound, however, there are minor inconsistencies with different cell backgrounds used for different assays (ie. CHO-K1 for spectroscopy v HEK293T for Western blots), which may influence the results.

Specific comments:

1. Figure S1 was difficult to understand and I found the figure legend confusing. Is b) the cAMP response in CHO cells? Is d) the cAMP response in HEK293T cells? Could you please add titles to the graphs so it is easier to see what results are from each cell line.

We apologize for not being clear on the figure legends in Fig. S1. The cAMP response in (b) was in CHO-K1 cells and in (d) was in HEK293T cells. In order to visualise the cAMP response for isoproterenol titration we performed a concentration series experiment with isoproterenol on the constructs as shown in Figure R2 (see above). Figure R2 a,b and d were performed in CHO-K1 cells and Figure R2 c was performed in HEK293T cells. **Fig. S1** in the supplementary information of the manuscript (same as R2 shown above) has been updated in the revised manuscript.

2. I could not find the methods for the cAMP fluorometric assay in the methods section. Could you please check that this is in the methods section. And if it is not, could you please add it to the methods section?

We apologize. The methodology of the cAMP fluorometric assay has been added to the Materials and Methods section (page 11 para 5) of the revised manuscript.

3. Why are there 2 standard curves for the cAMP assay (Figure S1a and S1c)?

The two standard curves in original Fig. S1a, c were from two independent experiments performed on two different days. As reviewer 1 has asked for more meaningful titration series for the effect of isoproterenol on different constructs we have now fully revamped **Fig. S1** in the revised manuscript.

4. Why do the two standard curves have different Emax (200cnts vs 40 cnts)?

The different values of Emax of the standard curves were due to them being performed on different days and using different detector sensitivity in the plate reader for each measurement. We apologize for not clearly explaining this. Note that this data is no longer shown in the revised version of **Fig. S1** as explained above.

5. Compared to wild type in Fig S1b, it looks like the IL3 tag reduces the basal constitutive activity of the b2AR compared to the WT (80 cnts for IL3 vs 55 cnts for WT), which is a difference. Also the net cAMP response looks different by eye for the IL3 construct compared to the WT (80-15 = 65 cnts for IL3 vs 55-10 = 45 cnts for WT). Could the authors please explain why they think that there are no functional differences between the differently tagged receptors, when this appears to be a functional difference?

We now measured a concentration series for each construct and normalised the fluorescence intensity to see the trend of agonist activation which stays the same between the wildtype and each construct (See Figure R2 above, and **Fig. S1** and **Table S1** in the revised manuscript).

6. I can't find the statistics to support the conclusion by the authors that the cAMP responses are unaltered. Could you please do the statistical tests on the data presented in Figure S1?

We apologize. We now provide three independent experiments for each concentration series of the construct (**Fig. S1**).

7. Table S1 shows the interpolated cAMP concentrations. The cAMP response in the presence of isoproterenol at the IL3 (15 cnts), S (10 cnts) and WT (12 cnts) in figure S1b seem to be outside the linear range of the cAMP standard curve shown in figure S1a, therefore they may be unreliable. Could the authors please check that these values are indeed within linearity? If not, the assays would need to be repeated to ensure all data points are within the linear range of the assay.

The data concerning table S1 have been updated now based on the titration experiments mentioned before so that concern no longer applies to the experiments shown.

8. For Figure 2/figure S8 why were the Western blots done with HEK293T cells, when the cAMP and diffusion experiments for the NT b2AR were done in a CHO cell background? These should have been done in the same cell background, since the cell background could influence the trafficking/mobility of the receptors. Could you please show the same Western blot data for the receptors expressed in the CHO cell background.

We have performed now additional experiments and show them in the supplementary information of the revised manuscript (western blot with CHO-K1 cells, **Fig. S10**). We initially performed the western blots with only HEK293T cells as they expressed more protein relative to the CHO-K1 cells used in time resolved fluorescence. We agree with the reviewer that this is an inconsistency in this study. As a solution we increased the expression levels in the CHO-K1 cells by transfecting it with double the amount of vector DNA and giving it more time post transfection (48 hr) before cell lysis and fractionation. We compare both HEK293T and CHO-K1 expression on the same blot with staining for both G β and β_2 AR (Figure R5a below). Despite unspecific binding, we can discern the EGFP bound β_2 -AR bands in both HEK293T cells and CHO-K1 cells expressing NT. The absence of signal in the cytoplasm fraction shows that no free receptor is present in the cytosol of both HEK293T and CHO-K1 cells as expected. The blot with anti-G β staining (Figure R5a, right) shows less amount of receptors in CHO-K1 cells than in HEK293T cells, despite the overall higher protein content of the former. This makes CHO-K1 cells highly useful for FCS where low degree of expression is favourable.

The blot with anti-EGFP staining (Figure R5b, left) also shows a similar trend with no cytosolic expression of EGFP. In addition, CHO-K1 cells shows bands for both non-transfected and transfected CHO-K1 cells which we attribute to unspecific binding. The anti-G β staining (Figure R5b, right) shows that the protein amount loaded between HEK293T cells and CHO-K1 cells are similar (Figure R5a) and leads to the previous conclusion that CHO-K1 cells express lower amount of fluorescent receptors compared to HEK293T cells. Note that in the case of Figure R5 a, right and b, left and right the gel was cut and reimaged at a longer exposure time without the positive control or the HEK93T cell samples to

account for the lower expression which gave rise to a relatively fainter signal for the CHO-K1 cell samples. The lane for pure GFP protein is not shown in the case of anti-G β staining (Figure R5b, right).

Figure R5. HEK293T cells and CHO-K1 cells were transiently transfected with NT and the whole lysate (W), the cytosolic (C) and the membrane fraction (M) were blotted against anti- β_2 -AR (a, left) and anti-GFP antibody (b, left). Right panels show the same blot against G β , which served as loading control. Please note that the GFP lane is not shown here in the anti-G β staining (b, right) as the purified protein does not show a band for G β . The positive control for the anti- β_2 -AR antibody was the membrane fraction of HEK293T cells transiently transfected with wild type β_2 -AR. As a positive control for the anti-GFP antibody we used purified GFP. Expected band sizes are listed on the right and respectively marked on the blots for easier orientation.

9. Since the S receptor appears to be behaving differently, did the authors do the Western blots for the S receptor? How did this receptor compartmentalize compared to the NT receptor?

The western blot for the S receptor on both HEK293T cells and CHO-K1 cells have now been added in the supplementary information of the revised manuscript as **Fig. S11**.

From Figure R6 we see that for the blot with anti- β_2 -AR staining (Figure R6a, left), the bands for β_2 -AR do not show up on the cytosol but there is visible unspecific binding for the antibody similar to the blot with anti- β_2 -AR staining for cells expressing NT. The anti-G β staining (Figure R6a, right) shows that the protein amount loaded for the CHO-K1 samples is more relative to HEK293T samples.

The blot with the anti-SNAP tag staining (Figure R6b, left) shows the bands for SNAP bound β_2 -AR and no free protein in the cytosol. The positive control for the SNAP tag protein shows multiple bands, along with the expected band at 20 kDa. This is due to the purified protein being produced as a fusion of SNAP tag fused to MXE-CBD complex (from manufacturer, NEB). So both SNAP-MXE-CBD and

SNAP-MXE show up as bands. The lane for pure SNAP tag protein is not shown in the case of anti-G β staining.

10. Can the S receptor internalize with the organic fluorophore bound? And in particular does it internalize when stimulated with ISO?

To answer this important question, we performed confocal imaging of CHO-K1 cells, now shown in the new **Fig. S8** (Figure R7 below). The CHO-K1 cell expressing S shows the presence of fluorescent receptors inside the cell which serves as evidence that receptors can internalize even when bound to organic fluorophore. Moving with the previous studies^{16,17} that receptors do internalise upon activation we would expect the S receptor to be internalized upon agonist activation.

In addition, the possibility of free dye in the cytoplasm or outside the cell causing the fast diffusion constant might be excluded as the washing steps usually removes any unbound dye. In addition, the diffusion constant expected from free dye in an aqueous solution would be $\sim 400 \mu\text{m}^2\text{s}^{-1}$ ^{18,19}, an order of magnitude higher than what we observe from S transfected and labeled cells. The text in the revised manuscript has been updated with the changes (page 5).

References

- 1 Scott, M. G. H., Swan, C., Jobson, T. M., Rees, S. & Hall, I. P. Effects of a range of β 2 adrenoceptor agonists on changes in intracellular cyclic AMP and on cyclic AMP driven gene expression in cultured human airway smooth muscle cells. *Br J Pharmacol* **128**, 721-729, doi:10.1038/sj.bjp.0702829 (1999).
- 2 Lakowicz, J. R. in *Principles of Fluorescence Spectroscopy* (ed Joseph R. Lakowicz) 353-382 (Springer US, 2006).
- 3 Kravets, E. *et al.* Guanylate binding proteins directly attack Toxoplasma gondii via supramolecular complexes. *eLife* **5**, e11479, doi:10.7554/eLife.11479 (2016).
- 4 Möller, J. *et al.* Single-molecule analysis reveals agonist-specific dimer formation of μ -opioid receptors. *Nature Chemical Biology*, doi:10.1038/s41589-020-0566-1 (2020).
- 5 von Kleist, L. *et al.* Role of the Clathrin Terminal Domain in Regulating Coated Pit Dynamics Revealed by Small Molecule Inhibition. *Cell* **146**, 471-484, doi:10.1016/j.cell.2011.06.025 (2011).
- 6 Dutta, D., Williamson, C. D., Cole, N. B. & Donaldson, J. G. Pitstop 2 Is a Potent Inhibitor of Clathrin-Independent Endocytosis. *PLOS ONE* **7**, e45799, doi:10.1371/journal.pone.0045799 (2012).
- 7 Day, C. A. & Kenworthy, A. K. Tracking microdomain dynamics in cell membranes. *Biochimica et Biophysica Acta (BBA) - Biomembranes* **1788**, 245-253, doi:10.1016/j.bbamem.2008.10.024 (2009).
- 8 Honigmann, A. *et al.* Scanning STED-FCS reveals spatiotemporal heterogeneity of lipid interaction in the plasma membrane of living cells. *Nature Communications* **5**, 5412, doi:10.1038/ncomms6412 (2014).

- 9 Koukalová, A. *et al.* Lipid Driven Nanodomains in Giant Lipid Vesicles are Fluid and Disordered. *Scientific Reports* **7**, 5460, doi:10.1038/s41598-017-05539-y (2017).
- 10 Kure, J. L., Andersen, C. B., Mortensen, K. I., Wiseman, P. W. & Arnsparng, E. C. Revealing Plasma Membrane Nano-Domains with Diffusion Analysis Methods. *Membranes* **10**, doi:10.3390/membranes10110314 (2020).
- 11 Calebiro, D. *et al.* Single-molecule analysis of fluorescently labeled G-protein-coupled receptors reveals complexes with distinct dynamics and organization. *Proceedings of the National Academy of Sciences* **110**, 743, doi:10.1073/pnas.1205798110 (2013).
- 12 Saffman, P. G. & Delbrück, M. Brownian motion in biological membranes. *Proceedings of the National Academy of Sciences of the United States of America* **72**, 3111-3113, doi:10.1073/pnas.72.8.3111 (1975).
- 13 Hughes, B. D., Pailthorpe, B. A. & White, L. R. The translational and rotational drag on a cylinder moving in a membrane. *Journal of Fluid Mechanics* **110**, 349-372, doi:10.1017/S0022112081000785 (1981).
- 14 Petrov, E. P. & Schwille, P. Translational Diffusion in Lipid Membranes beyond the Saffman-Delbrück Approximation. *Biophysical Journal* **94**, L41-L43, doi:10.1529/biophysj.107.126565 (2008).
- 15 Sungkaworn, T. *et al.* Single-molecule imaging reveals receptor-G protein interactions at cell surface hot spots. *Nature* **550**, 543-547, doi:10.1038/nature24264 (2017).
- 16 Shumay, E., Gavi, S., Wang, H.-y. & Malbon, C. C. Trafficking of β 2-adrenergic receptors: insulin and β -agonists regulate internalization by distinct cytoskeletal pathways. *Journal of Cell Science* **117**, 593, doi:10.1242/jcs.00890 (2004).
- 17 Kim, H., Lee, H. N., Choi, J. & Seong, J. Spatiotemporal Characterization of GPCR Activity and Function during Endosomal Trafficking Pathway. *Analytical Chemistry*, doi:10.1021/acs.analchem.0c03323 (2021).
- 18 Petrov, E. P. & Schwille, P. in *Standardization and Quality Assurance in Fluorescence Measurements II: Bioanalytical and Biomedical Applications* (ed Ute Resch-Genger) 145-197 (Springer Berlin Heidelberg, 2008).
- 19 Petrášek, Z. & Schwille, P. Precise Measurement of Diffusion Coefficients using Scanning Fluorescence Correlation Spectroscopy. *Biophysical Journal* **94**, 1437-1448, doi:10.1529/biophysj.107.108811 (2008).

REVIEWERS' COMMENTS:

Reviewer #1 (Remarks to the Author):

Although I had originally raised concerns about how this work would impact the biological relevance of what was measured, the authors have responded in an appropriate and forthright way.

Reviewer #2 (Remarks to the Author):

I am happy with the revised version of the manuscript. All my concerns were addressed.

Sincerely yours,
Lucas

Reviewer #3 (Remarks to the Author):

Summary:

The aim of the study is to characterise the movement of the b2AR in the plasma membrane. The study was conducted using GFP or SNAP/TAG b2AR, with tags in different positions. Movement of the b2AR was assessed using fluorescence correlation spectroscopy, time-resolved anisotropy and polarised fluorescent correlation spectroscopy to determine the receptor mobility. The conclusions of the study are that different pools of b2AR exists within plasma and intracellular membranes that have different mobility properties.

Comments:

Thank you to the authors for the additional experiments included in the revised manuscript. Most of my concerns/questions have been adequately addressed by the authors. Only one of my concerns was not adequately addressed:

The statement at the end of paragraph 1 of the results states that the cAMP responses between the different receptors are "unaltered". I would consider changing the word "unaltered" to something more conservative. The results in Figure S1b show a difference in the cAMP concentration-response curves for WT v IL3 and the EC50 values in Table S1 show an EC50 of 1.3+/-0.8 nM for WT but 32.9+/-19.5, which represents quite a large shift. However, the error on the IL3 data seems quite large, which probably makes this statistically insignificant.

Could you please include the number of experiments in the figure S1 legend.

For Table S1, are the errors shown as SD or SEM? Please add this information to the Table legend. It is important for assessing the statistical significance of the differences.

Could you also please provide statistics on this table to show that the results are not statistically different from one another. And add to the table legend what statistical test you used to compare the EC50 values.

Dear Reviewers,

We thank you for your conditional acceptance of our work. We have carefully revised the parts, which were not adequately addressed in the last resubmission. Please find all answers in green and the respective text changes in the main manuscript indicated in blue.

Reviewer #3's comment:

Thank you to the authors for the additional experiments included in the revised manuscript. Most of my concerns/questions have been adequately addressed by the authors. Only one of my concerns was not adequately addressed:

1) The statement at the end of paragraph 1 of the results states that the cAMP responses between the different receptors are "unaltered". I would consider changing the word "unaltered" to something more conservative. The results in Figure S1b show a difference in the cAMP concentration-response curves for WT v IL3 and the EC50 values in Table S1 show an EC50 of 1.3+/-0.8 nM for WT but 32.9+/-19.5, which represents quite a large shift. However, the error on the IL3 data seems quite large, which probably makes this statistically insignificant.

We thank the reviewer for pointing this out, and have now rephrased the sentence (page 4 para 1).

2) Could you please include the number of experiments in the figure S1 legend.

We apologize for not mentioning this before. The data for each ligand concentration comes from three independent experiments. The legend for **Supplementary Figure 1** (page 2, Supplementary information) and **Supplementary Table 1** (page 24, Supplementary information) has been updated in the revised manuscript.

3) For Table S1, are the errors shown as SD or SEM? Please add this information to the Table legend. It is important for assessing the statistical significance of the differences.

The errors shown in Supplementary Table 1 are the standard error of the mean from fitting with eq. 1. Supplementary Table 1 legend has been updated in the revised manuscript (page 24, Supplementary information).

4) Could you also please provide statistics on this table to show that the results are not statistically different from one another. And add to the table legend what statistical test you used to compare the EC50 values.

A two-sample t-test was performed between each construct and its corresponding WT for statistical significance. All constructs except TAG are not statistically significant for its corresponding WT. In the case of TAG, the fit with eq. 1 gave a R² value of 0.94 (Figure R1a) relative to 0.99 for WT (HEK293T) (Figure R1b) and both constructs are saturated at 100 nM isoproterenol. Hence, they exhibit a similar trend in their activation behaviour comparable to the other constructs and the respective WT. The supplementary information has been updated in the revised manuscript (page 24, Supplementary Table 1).

Figure R1. cAMP response curves of TAG (a) and WT (HEK293T) (b) measured using a fluorimetric cAMP assay.